

# Submesoscale dispersion of surface drifters in a coastal sea near offshore wind farms

Ulrich Callies[1], Ruben Carrasco[1], Jens Floeter[2], Jochen Horstmann[1], and Markus Quante[1]

[1]Helmholtz-Zentrum Geesthacht, Max-Planck-Str. 1, 21502 Geesthacht, Germany
[2]Institut für Hydrobiologie & Fischereiwissenschaft, Universität Hamburg, Germany

*Correspondence to:* U. Callies (ulrich.callies@hzg.de)

**Abstract.** We analyse relative dispersion of surface drifters released as pairs (6 instances) or triplets (2 instances) during three field experiments in the German Bight in close proximity to wind farms. Drifter pairs can be classified in a remarkably clear way into those with spatial separation growing either exponentially or non-monotonously. There is some tentative evidence that exponential relative dispersion growth rates preferably occur for drifter pairs that are most exposed to the possible influence of a wind farm. Kinetic energy spectra and velocity structure functions suggest that turbulent energy could be injected by tides, possibly also via an interaction between tidal currents and wind turbine towers. Applicability of inertial range turbulence theory, however, can be doubted given distinct peaks of overtides observed in velocity power spectra. More comprehensive studies would be needed to better separate submesoscale effects of wind farms, tides and possibly baroclinic instabilities on observed drifter behaviour in a complex coastal environment.

## 1 Introduction

Observing the spreading of drifters deployed pairwise is a powerful tool for analysing submesoscale flow structures. Submesoscale features are of interest for different reasons. From a theoretical point of view, understanding mesoscale turbulent features helps understand the mechanism how energy in a 2D quasi-geostrophic regime cascading towards larger scales (inverse energy cascade, see Charney, 1971) can nevertheless lose geostrophic balance and be injected to the microscale, where it is dissipated (McWilliams, 2008). Another reason is a more practical one. Knowing about the efficiency of relative dispersion at the submesoscale is important for proper simulation of early phase spreading of pollutant patches. It is crucial to know whether spreading will be driven by mesoscale structures resolved in numerical models (non-local dispersion) or if submesoscale turbulence on the scale matching the size of a pollutant patch is energetically relevant (local dispersion). In the latter case growth of a small-size oil slick, for instance, will exceed the rate predicted by traditional parametrizations in terms of hydrodynamic currents resolved in a model (Özgökmen et al., 2012).



In this study we analyse drift trajectories in the German Bight (North Sea) that cover just short periods (maximum 3.9 days). The German Bight (Fig. 1) is characterized by frequent eddies and meanders on different scales. Nearshore gyres may occur or be absent depending on prevailing wind conditions or baroclinic instabilities in connection with fronts (Becker et al., 1992), for instance. Focussing on local conditions distinguishes our study from others that consider drifters spreading over larger spatial

scales (e.g. Corrado et al., 2017). Initial separations of drifter pairs we analyse are much below the local internal radius of deformation, which in the German Bight is in the range of approximately 2-20 km (Becker et al., 1983, 1999; Badin et al., 2009). Therefore our experiments explore the sub-mesoscale regime in which geostrophic horizontal turbulence interacts with vertical mixing (e.g. McWilliams, 2008), possibly triggered by the presence of wind farms (Floeter et al., 2017). Departure from geostrophic dynamics in submesoscale eddies can be quite substantial (Ohlmann et al., 2017).

A recent summary of relative dispersion in the ocean was given by Corrado et al. (2017). Analysing data from the Global Drifter Program, theses authors found consistent behaviours in different ocean sub-basins. Conditions in coastal regions, however, are generally less homogeneous than in the open sea and may give rise to variable flow features that vary substantially on a scale of only few kilometres (Ohlmann et al., 2012). In the German Bight, strong tidal waves ($M_2$) become distorted and shallow-water overtides ($M_4$ and $M_6$) are generated via reflection and non-linear transformation processes (Stanev et al.,

2014, 2016). The German sector of the German Bight is also an area in which a large number of offshore wind farms (OWFs) are planned, built or already operated. Although generation of turbulent wakes by OWF structures is a known effect (e.g. Li et al., 2014), the number of targeted studies of the impacts of OWFs on hydrodynamic conditions is very limited. Impacts on hydrodynamic conditions might occur via either changes in the atmospheric wind field or tidally induced mixing in an array of wind farm foundations. Seasonal variation of stratification is a crucial factor influencing the North Sea food web (e.g. Ruardij

et al., 1997). While Carpenter et al. (2016) estimated little impact of OWFs on mean stratification in the German Bight, Floeter et al. (2017) found some observational evidence that stirring effects might increase vertical mixing and create upwelling cells near the wind farms.

Based on data from experiments in the Mediterranean Sea, Schroeder et al. (2011) raised doubts that sub-mesoscale turbulent eddies are pervasive phenomena underlying turbulent transports. Alternatively, turbulent transports may be governed by larger

mesoscale hydrodynamic features. Such non-local transports (or drifter dispersion) are expected to occur in combination with Eulerian energy wave number spectra $\propto k^{-3}$ or steeper (Bennett, 1984). Kraichnan (1967) predicted this for the enstrophy-cascading inertial range of 2D-turbulence, for instance. Indicative of a non-local regime driven by flow features larger than drifter separation is exponential growth of relative drifter dispersion (LaCasce, 2008). By contrast, local dispersion with power law dependence on time should coincide with a shallower slope of the energy spectrum, indicating the presence of energetic

small scale eddies. Özgökmen et al. (2012) compiled relevant analyses available at that time (LaCasce and Ohlmann, 2003; Koszalka et al., 2009; Lumpkin and Elipot, 2010; Berti et al., 2011; Schroeder et al., 2011), more recent studies were reported by Beron-Vera and LaCasce (2016), Corrado et al. (2017), Poje et al. (2017) or Sansón et al. (2017). An assessment of the influence of the different flow regimes on turbulent transport is complicated by the fact that exponential increase of tracer separation in time is also characteristic of so-called Lagrangian chaos dealt with in the dynamical systems theory. This occurs

when passive objects show chaotic movements sensitive to initial conditions although they are embedded into laminar Eulerian





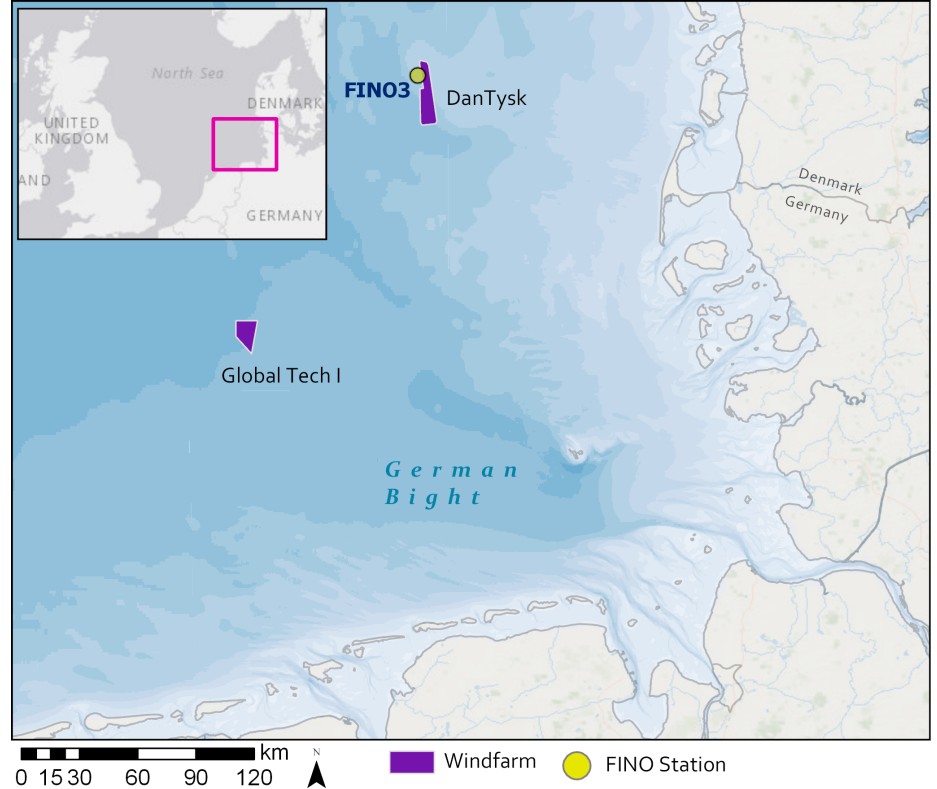

**Figure 1.** The study area German Bight. Drifter experiments were conducted in the vicinity of the two wind farms indicated in the plot. Research station FINO3 provides hydrodynamic currents on a 10 min basis.

currents (Boffetta et al., 2000; Tsinober, 2001, Sections 4.2, 4.3). Wiggins (2005) reviews applications of the dynamical systems approach in the context of oceanography.

The issue of either local or non-local dispersion at submesoscale seems not yet to have been solved. Berti et al. (2011) found early phase exponential separation at scales of the order of 1 km. By contrast, Corrado et al. (2017) observed rates of dispersion

5    at the submesoscale being about one order of magnitude higher than at the meso- or largescale and took this as an indication that dispersion was increased by the action of local eddies similar in size to drifter separation. In an experiment specifically targeted to a persistent coastal buoyant front possibly containing sub-mesoscale mixed layer instabilities, also Schroeder et al. (2012) found indications of relative dispersion enhanced by local dynamics.

The data studied here represent quite a complex situation in which effects of tides modified by travelling under shallow

10   sea conditions, baroclinic instabilities on the scale of the Rossby deformation radius and anthropogenic effects of wind farms may possibly combine. Sect. 2 describes the data available, the method of spectral analysis we applied to drifter velocities and summarizes basic concepts of two-particle statistics. In addition it explains how simulated counterparts of observed trajectories





were produced. The results section starts with a detailed analysis of observed drifter trajectories and drifter pair separations (Sect. 3.1). Observed trajectories influenced by changing weather conditions are supplemented with corresponding simulations. Sect. 3.2 presents spectral analyses of both Eulerian and Lagrangian current velocities. Sect. 3.3 deals with two-particle statistics like separation velocities and velocity structure functions. Finally, Sect. 3.4 presents examples of simulated drifter

dispersion based on two different stochastic parametrizations. After a discussion of our findings in Sect. 4, conclusions are summarized in Sect. 5.

## 2   Material and methods

### 2.1   Observational data

Surface drifter data were collected during three research cruises with RV Heincke (HE445, HE490, HE496) in the German

Bight in the years 2015 and 2017. Table 1 summarizes for all drifters positions and times of their deployment. In addition, the table provides lengths of drifter tracks together with the linear distances between initial and final locations. We used drifters of type MD03i from Albatros Marine Technologies, shaped as cylinders with 0.1 m diameter and 0.32 m length. About 0.08 m protrude from the water surface, the ratio of drag area in to drag area outside the water is 33.2. Drogues of 0.5 m both length and diameter are attached 0.5 m below the sea surface, so that drifters are supposed to reliably represent currents in a surface

layer of about 1 m depth. No drogue presence sensors were mounted for checking the conditions of the devices.

    Drifter positions were obtained from the Global Positioning System (GPS) and transmitted to the lab via the satellite communication system Iridium. A lab test was set up to evaluate accuracies of GPS devices. Four drifters were deployed in a small water tank at fixed positions, so that changes of their distances relative to each other (recorded for each of six pairs yielded from the four drifters) could directly be attributed to errors of GPS based localization. Based on 48 hours of observations,

the $50^{th}$, $90^{th}$ and $95^{th}$ percentiles of relative distance errors were 12.4, 33 and 42 meters, respectively. Real errors could be slightly larger because our test could not take into account possible effects of orbital motions due to waves.

    In all field experiments sampling rates were about once every 20 min. For being able to calculate time dependent separations between drifters, all drifter locations were linearly interpolated to regular 20 min time intervals. Drifter velocities were derived from these interpolated regular data.

**Drifter set A:** On 21 May 2015, three drifters ($A_2$ - $A_4$) were deployed as a triplet near the wind farm DanTysk (Fig. 2a). DanTysk covering an area of roughly $19 \times 5$ km$^2$ is located about 70 km to the west of the offshore coastal islands near the Danish/German border (Fig. 1). The three drifters crossing the area of the wind farm were tracked for a maximum time of 3.7 days (see Table 1).

    The three drifters are a subset of nine drifters released in May 2015 during a longer cruise (HE445) of RV *Heincke*.

The other six drifters were released individually and monitored between 9 and 54 days while they were drifting across the German Bight. Their tracks were analysed by Callies et al. (2017). Here, just drifter $A_5$ will be used, analysing





its Lagrangian velocity power spectrum (see Sect. 3.2). Data from all nine drifters are freely accessible from a data repository (Carrasco and Horstmann, 2017).

**Drifter set B:** On 29 June 2017, one drifter triplet ($B_1$, $B_2$ and $B_3$) was deployed to the west of wind farm Global Tech I (Fig. 4a). The wind farm (Fig. 1) comprises 80 turbines with tripod foundations. It covers an area of about $41$ km$^2$ and

is located more than 90 km to the north-west of the German island Juist. Water depth in the region is about 40 m. Drifter data taken on cruise HE490 of RV *Heincke* are freely accessible at Carrasco et al. (2017a).

Drifters were tracked for 1.9, 2.9 and 3.9 days, respectively. Another pair of drifters ($B_4$ and $B_5$) was deployed within the wind farm about five hours later. These drifters were tracked for 1.9 and 2.9 days, respectively (see Fig. 4d).

**Drifter set C:** On 14/15 September 2017, five drifter pairs were deployed with spatial spacing of 5 nautical miles along a

north-south transect to the west of wind farm Global Tech I (Fig. 6a). Drifter tracks were recorded for up to 3.5 days. For drifters $C_9$ and $C_{10}$ some technical problems encountered after drifter deployment endured for nearly one day. Fig. 6 shows only data after these problems had been settled and signals were obtained on a regular basis. All data taken on cruise HE496 are freely accessible at Carrasco et al. (2017b).

Note that all launch locations and times listed in Table 1 refer to the first signal received from the positioning system. As a

result, initial distances seem larger than they actually were at the time of drifter deployment, which may have taken place about 30 min before.

Eulerian surface currents observed at 2 m depth were available from research station FINO3 (https://www.fino3.de), located approximately 80 km off the German coast in the immediate vicinity of the wind farm DanTysk where experiment A took place (Fig. 1). Time resolution of these measurements taken with an acoustic Wave and Current Profiler (AWAC) is 10 min.

For technical reasons each hour one of these measurements is usually skipped. A special period without such data gaps (April-May 2010), needed for spectral analysis, did unfortunately not overlap with the time periods of our drifter experiments.

## 2.2 Spectral Analyses

Power spectra of both Eulerian and Lagrangian drifter velocities have been calculated using the maximum entropy method (MEM) based on algorithms presented in Marple (1987) and Press et al. (2002). This method has been chosen, since it is

very efficient in detecting narrow spectral features or sharp peaks even if the underlying data series have a quite low number of sample points ($N$). The behaviour of the spectral estimate using MEM depends on the appropriate choice of the order of the autoregressive model ($M$). The number of peaks typically increases with $M$. If the order is chosen too high, spurious peaks may occur in the spectra. Therefore, several spectra for each case with different model orders have been calculated. The model order suggested by the Akaike information criterion (Akaike, 1974) has been found to be too low, known peaks were

not resolved. Here an order selection of $N/4$ to $N/3$ produced satisfactory results. For some cases (longer data series) MEM spectra have been compared to FFT based power spectra to verify main peaks and spectral slopes as they are discussed here.





## 2.3 Velocity increments and structure functions

Let $D_{ij}(t)$ denote separation between two drifters $i$ and $j$, at time $t$ being located at $\boldsymbol{x}^{(i)}(t)$ and $\boldsymbol{x}^{(j)}(t)$, respectively:

$$D_{ij}(t) = \mid \boldsymbol{x}^{(i)}(t) - \boldsymbol{x}^{(j)}(t) \mid \tag{1}$$

Given a cloud of drifters, the mean squared separation of $N$ pairs of drifters provides a measure of relative two-particle
dispersion

$$D^2(t) = \langle D_{ij}^2(t) \rangle = \frac{1}{N} \sum_{i \neq j} D_{ij}^2(t) \tag{2}$$

where brackets denote averaging over all particle pairs. In the present study, however, we will analyse each drifter pair separately, so that squared separation $D_{ij}^2(t)$ will be our key parameter. The reason for doing so is that combining drifter pairs would make results less transparent and more difficult to discuss. Of course such detailed analysis would not be feasible if the
number of drifter pairs studied would be larger.

    Relative dispersion is to be distinguished from absolute dispersion, a parameter from single particle statistics that describes a particle cloud's spread around its center of mass in combination with its drift from its release point. Differences between absolute and relative dispersion are relevant at medium time scales when two particle velocity cross correlation depends on the character of Eulerian flows (LaCasce, 2008). Being the second moment of the distribution of relative particle displacements,
relative dispersion is informative when this distribution is nearly Gaussian. Otherwise studying full distributions of relative drifter separations may be preferable (LaCasce, 2010).

    In his seminal paper, Richardson (1926) assumed that separation of particle pairs will hardly be affected by eddies larger in diameter than the distance between the two tracer particles. Similarly, turbulent structures much smaller than drifter separation will not much contribute to further spreading. A disadvantage of relative dispersion $D^2(t)$ is that its value at given time $t$ does
not necessarily relate to a specific spatial scale. Drifter pairs contributing to the average may travel under different flow regimes and thereby give rise to scale interference (Corrado et al., 2017). Considering Eulerian velocity differences $\delta \boldsymbol{v}^{(E)}(\boldsymbol{r}, t) = \boldsymbol{v}(\boldsymbol{x} + \boldsymbol{r}, t) - \boldsymbol{v}(\boldsymbol{x}, t)$ between two locations separated by distance $\boldsymbol{r}$ helps address the role of spatial scales. If possible implications of non uniform sampling due to specific flow structures are neglected (Poje et al., 2017), Eulerian velocities can be identified with Lagrangian drifter velocities. As a convenient scalar parameter the following Eulerian longitudinal velocity difference (Poje
et al., 2014, 2017) can be used,

$$\delta v_{\parallel}^{(E)}(r, t) = \delta \boldsymbol{v}^{(E)}(\boldsymbol{r}, t) \cdot \hat{\boldsymbol{r}}(t) \tag{3}$$

with $r = \parallel \boldsymbol{r} \parallel$ and $\hat{\boldsymbol{r}} = \boldsymbol{r}/r$. In 3D turbulence, the corresponding transverse velocity difference $\delta \boldsymbol{u}_{\perp}^{(E)}$ could have any direction within a plane perpendicular to $\hat{\boldsymbol{r}}$ (e.g. Lévêque and Naso, 2014). In 2D, however, its orientation is well defined and the component can be obtained as

$$\delta v_{\perp}^{(E)}(r, t) = \delta \boldsymbol{v}^{(E)}(\boldsymbol{r}, t) \times \hat{\boldsymbol{r}} \tag{4}$$



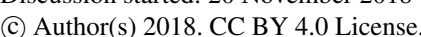


where the convenient 2D analog $\boldsymbol{a} \times \boldsymbol{b} = a_x b_y - a_y b_x$ of the 3D vector product was used. The second-order structure function is defined as the second moment of velocity differences between two neighbouring points (e.g. Kolmogorov, 1941; Pope, 2000). Based on Eqs. (3) and (4) it can again be decomposed into longitudinal and transverse components (Poje et al., 2017):

$$S_{2,\|}^{(E)}(r) = \langle \{\delta v_\|(r,t)\}^2 \rangle \quad ; \quad S_{2,\perp}^{(E)}(r) = \langle \{\delta v_\perp(r,t)\}^2 \rangle \tag{5}$$

In Eq. (5) we assumed isotropic conditions so that vector $\boldsymbol{r}$ can be replaced by its scalar length $r$. Angular brackets denote averaging over each subset of paired locations after the full data were binned with regard to distance $r$ (regardless of time $t$).

Both time evolution of relative dispersion $D^2(t)$ and spatial scale dependence of velocity structure functions like $S_{2,\|}^{(E)}(r)$ can be theoretically linked to wave number power laws that hold for turbulent kinetic energy. In two dimensions the spectrum of turbulent energy $E$ as function of wave number $k$ may combine an inverse energy cascade at large scale with a direct enstrophy

cascade at smaller scale, separated by a frequency where energy is injected (Kraichnan, 1967; Lesieur, 1997; LaCasce, 2008; Boffetta and Ecke, 2012):

$$E(k) \propto \begin{cases} \varepsilon^{2/3} k^{-5/3} & \text{inverse energy cascade} \\ \eta^{2/3} k^{-3} & \text{direct enstrophy cascade} \end{cases} \tag{6}$$

Here, energy dissipation $\varepsilon$ assumes the meaning of an energy flux to larger scales and $\eta$ denotes an enstrophy dissipation or transfer rate. The spectrum for the 2D inverse energy cascade is identical with that for the direct cascade towards smaller scales

that Richardson (1926) derived for 3D turbulence. From Eq. (6) the following explicit time dependences of squared drifter separation can be derived (Babiano et al., 1990; Ollitrault et al., 2005),

$$D^2(t) \propto \begin{cases} \varepsilon t^3 & \text{inverse energy cascade} \\ \exp\left(c\eta^{1/3}t\right) & \text{direct enstrophy cascade} \end{cases} \tag{7}$$

with some positive constant $c$. It is known, however, that observing scaling laws (7) does not necessarily prove existence of an inertial energy cascade and the corresponding spectral power law (e.g. Zouari and Babiano, 1994; Tsinober, 2001).

After sufficiently long time particle motions will become decorrelated and the power law behaviour of squared drifter separation will settle into normal diffusion (Kraichnan, 1966) for which relative diffusivity is expected to be constant (LaCasce and Bower, 2000) and twice the value of absolute diffusivity considered by Taylor (1921).

Following K41 scaling (Kolmogorov, 1941), in the inertial range of two-dimensional turbulence one has (Babiano et al., 1985; Boffetta and Ecke, 2012):

$$S_{2,\|}^{(E)}(r),\ S_{2,\perp}^{(E)}(r) \propto \begin{cases} \varepsilon^{2/3} r^{2/3} & \text{inverse energy cascade} \\ \eta^{2/3} r^2 & \text{direct enstrophy cascade} \end{cases} \tag{8}$$

Eqs. (6) and (8) are special instances of a more general phenomenological correspondence between $E \propto k^{-\alpha}$ and $S_2^{(E)}(r) \propto r^{\alpha-1}$ for different values of $\alpha$. However, for steep spectra with $\alpha > 3$ this relationship does no longer hold and the velocity





structure function saturates at $r^2$ (Babiano et al., 1985). Boffetta and Ecke (2012) state that velocity structure functions may provide less information about small scale turbulent components than vorticity structure functions. The latter, however, are not available based on the drifter data of this study.

## 2.4 Drifter simulations

For drifter simulations we employed the 2D Lagrangian transport module PELETS (Callies et al., 2011), based on surface currents archived from the hydrodynamic model BSHcmod (Dick et al., 2001). BSHcmod is run operationally by the Federal Maritime and Hydrographic Agency (BSH). PELETS, developed at Helmholtz-Zentrum Geesthacht (HZG), is designed for particle tracking on unstructured triangular grids. If instead hydrodynamic fields are provided on a structured grid, as in the case at hand, introducing diagonals splits each rectangular grid cells into two triangles. Using a simple Euler forward method, particle velocities are updated each time a particle passes from one to another triangular grid cell. As a result of this concept, time step is not a constant. It has, however, an upper limit. If no edge is reached within 15 min, an additional update of drift velocity will be triggered.

BSHcmod is run on a two-way nested grid covering both the North Sea and the Baltic Sea. In the German Bight its horizontal resolution is 900 m. Although the vertical coordinate in BSHcmod is dynamical (Dick et al., 2008), re-gridded archived output represents surface currents in terms of the mean in an upper 5 m water column. Callies et al. (2017) found that an additional wind drag in terms of 0.6 % of the 10 m wind velocity $\boldsymbol{u}_{10m}$ is appropriate to compensate for the lack of vertical grid resolution in archived model output. Therefore, for simulating drifter location $\boldsymbol{x}$ as function of time $t$, the following equation is used:

$$\frac{d\boldsymbol{x}}{dt} = \hat{\boldsymbol{v}}_E = \boldsymbol{v}_E + \beta \boldsymbol{u}_{10m} \tag{9}$$

Here $\boldsymbol{v}_E$ denotes Eulerian marine surface currents from BSHcmod, archived on a 15 min basis, and $\boldsymbol{u}_{10m}$ atmospheric forcing from the regional model COSMO-EU (Consortium for Small-Scale Modelling; Schulz and Schättler, 2014) run by the German Meteorological Service (Deutscher Wetterdienst - DWD) with spatial resolution 7 km. The value 0.006 is assigned to weighting factor $\beta$.

As an option, in PELETS subscale turbulent processes can be included via a scale-dependent random diffusion term. Assuming that movements in the two dimensions are decoupled, updates of a particle's position vector $\boldsymbol{x}(t)$ after time $dt$ are described by the following discretized version of the corresponding stochastic Langevin equation for each vector component $x_i$:

$$dx_i(t) = x_i(t + dt) - x_i(t) = [\hat{v}_{E,i}(t) + v_i'(t)] \, dt = \hat{v}_{E,i}(t) dt + \sqrt{2K} \, dW(t) \tag{10}$$

The right hand side of this equation combines a deterministic Eulerian velocity component $\hat{v}_{E,i}(t)$ with a white-noise-driven diffusion term $v_i'(t)$. $K$ denotes horizontal eddy diffusivity and $W$ is a Wiener process, independent increments of which have a zero mean and a second order moment $\langle dW^2 \rangle = dt$. Eq. (10) is appropriate for time increments that exceed the time particles need to lose memory of turbulent momentum (Heemink, 1990; Zambianchi and Griffa, 1994). The assumption of a clear gap between scales of mean and turbulent motions, respectively, also underlies the common eddy-diffusion parametrization as the Eulerian analogue of the Lagrangian model Eq. (10).

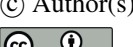



Following Schönfeld (1995), diffusivity $K$ is assumed to depend on a characteristic length scale $l$ according to a 4/3 power law (Stommel, 1949):

$$K(l) = K(l_0) \left( \frac{l}{l_0} \right)^{\frac{4}{3}} \tag{11}$$

For a reference length scale $l_0 = 1$ km we chose $K(l_0) = 1$ m²/s. This value roughly agrees with the value of 2.5 m²/s for a reference length scale of one nautical mile chosen by Schönfeld (1995). The length scale $l$ in Eq. (11) was chosen to equal spacing of the numerical grid.

To improve performance at early times after drifter deployment, Eq. (10) may be replaced by a random flight model that assigns a finite memory to turbulent drifter velocity (Durbin, 1980; Heemink, 1990; Griffa et al., 1995; LaCasce, 2008):

$$v_i'(t) = \left( 1 - \frac{dt}{T_L} \right) v_i'(t - dt) + \frac{\sqrt{2K}}{T_L} \, dW(t) \tag{12}$$

Here $T_L$ denotes the Lagrangian decorrelation time. For $dt = T_L$, Eq. (12) coincides with the turbulent component in Eq. (10). For drift times $t - t_0$ much exceeding $T_L$, the diffusivity $K$ equals the product $\sigma^2 T_L$ (Falco et al., 2000), where $\sigma^2$ denotes the turbulent velocity variance. With this substitution the random component of the turbulent velocity component $v_i'$ assumes the form $\sqrt{2\sigma^2/T_L} \, dW(t)$ which is more common (e.g. Griffa et al., 1995; Falco et al., 2000; Ohlmann et al., 2012). The advantage of Eq. (12) is that it directly refers to the scale dependent model parameter $K$ in Eq. (11).

# 3 Results

## 3.1 Drifter trajectories and separations

### 3.1.1 Drifter set A

Trajectories of drifters $A_2$, $A_3$ and $A_4$ are shown in Fig. 2a. Different colours are used to distinguish between periods with different wind conditions (Fig. 2b) or to highlight periods of special interest. Superimposed to tidal oscillations roughly oriented between south-east and the north-west, the drifter triplet first moves from the location of its deployment in the south-west of DanTysk towards the north-east, roughly in parallel with prevailing winds. Within about one day the drifters cross the wind farm area. After winds veered to blow from the north-west, residual transports reverse their direction and drifters cross the wind farm area once again. The third day is again characterized by winds from the south-west, giving rise to another reversal of the residual transport direction. Separation between drifters now becomes clearly noticeable on the scale of the plot. At the end of the observation period, winds change again and blow from the north-west. The reaction to this last change in wind direction, however, differs between drifters $A_2$ and $A_3$ (drifter $A_4$ has already been recovered at that time), reflecting gradients in residual current fields on a scale of few kilometres (the final distance between $A_2$ and $A_3$ is approximately 4 km).

Fig. 2c displays the simulated counterpart of trajectory $A_2$. Colour coding agrees with that used in Figs. 2a and b. Simulations well reflect the general patterns observed, which in particular confirms the reliability of winds underlying the simulations. On the mean, however, simulated transports are more southward, resulting in an error of about 8 km in the final locations



**Figure 2. (a)** Observed trajectories of drifters $A_2$, $A_3$ and $A_4$. **(b)** Wind conditions during the experiment. The panel also shows travel times of all trajectories. **(c)** Simulated trajectory $A_2$. Different colours partitioning the observational period are used consistently across all panels.



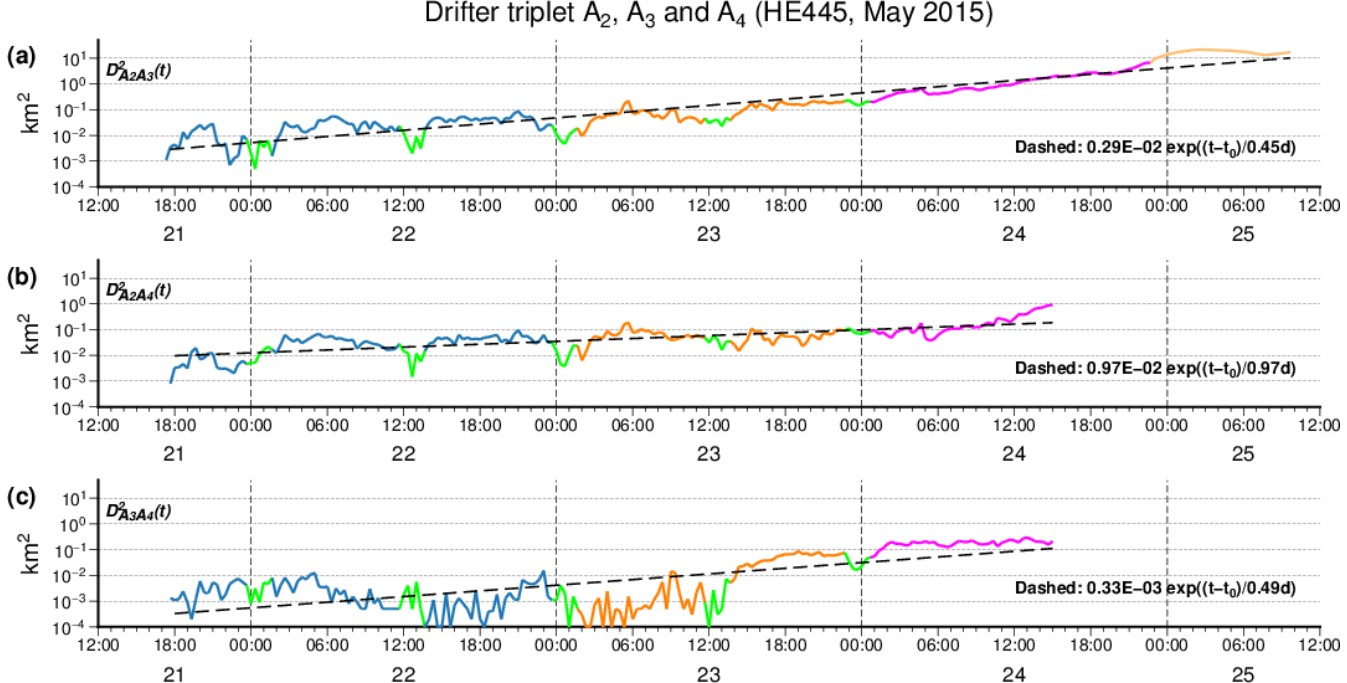

**Figure 3.** Time evolutions of pairwise distances between members of drifter triplet $A_2$, $A_3$ and $A_4$. Segmentation using different colours is consistent with Fig. 2. Dashed lines represent the fitted exponential growth models annotated in each graph.

predicted for drifter $A_2$ or $A_3$ (as initial location practically coincide, corresponding simulations differ just slightly in drift time). Simulation errors seem substantial relative to observed overall drifter displacements ($A_2$: 12.2 km; $A_3$: 8.1 km), but are very moderate in the light of the lengths of undulating drift paths ($A_2$: 87.4 km; $A_3$: 85.7 km; see Tab. 1).

For all three pairs yielded from the drifter triplet, semi-log plots in Fig. 3 show how squared drifter separations develop

with time. Techniques for the evaluation of three particle dispersion (LaCasce and Ohlmann, 2003; Berta et al., 2016) were not applied. Colours used for time segmentation are consistent with those in Fig. 2. For each drifter pair a model of exponential growth was fitted, possibly indicative of a non-local regime. For the two pairs $A_2$, $A_3$ and $A_3$, $A_4$ very similar e-folding times (about half a day) were found, corresponding with a bit less than one day for non-squared separation. For drifters $A_2$, $A_4$ the estimated e-folding time is approximately twice as large. It should be noted, however, that the fit in Fig. 3c is quite uncertain

and mainly based on the behaviour at larger distances. The more random behaviour observed at smaller distances might already reflect uncertainty in measurements (a squared distance of $10^{-3}$ km$^2$ corresponds with the $90^{th}$ percentile of errors measured in the lab, see Section 2.1). However, it is hard to tell why this uncertainty does not show up for the other two drifter pairs.

In Figs. 3a and b, dips (coloured in green) in separation between drifter $A_2$ and either $A_3$ or $A_4$ occur with regularity. According to Fig. 2a all those dips coincide with tidal currents oriented towards the south-east, possibly indicating convergent

surface currents related to gradients in bathymetry. The effect is evident mostly for the two pairs including drifter $A_2$ that tends




to separate from the other two drifters (see Fig. 2a). In Fig. 3c the tidal signal becomes clear only when separation of drifters $A_3$ and $A_4$ exceeds a value of approximately 100 m (corresponding with a squared separation of $10^{-2}$ km$^2$).

### 3.1.2 Drifter set B

This experiment comprised two drifter releases at slightly different locations. One triplet ($B_1$, $B_2$ and $B_3$) was released in the
west of wind farm Global Tech I (see Fig. 1) and drifters were tracked for between 1.9 and 3.9 days (see Tab. 1). Observations are shown in Fig. 4a, a corresponding simulation in Fig. 4b. With a delay of a bit more than five hours, another two drifters ($B_4$ and $B_5$) were deployed inside the wind farm and tracked for 1.9 and 2.9 days, respectively. Observations and a corresponding simulation are shown in Figs. 4d and e, respectively.

Time evolutions of squared drifter separations are presented in Fig. 5. For all drifter pairs an exponential model fitted to
the data revealed approximately the same e-folding time of half a day. However, again these fits must not be overrated as observations show large variability at small drifter distances. Spatial scales at which such variability occurs seem comparable to those in experiment A (compare Figs. 5d and 3c) and again fluctuations might be attributable to uncertainties in GPS based drifter localization. On the other hand, variations show a certain coherence in time and sometimes include distances (up to 300 m, see Fig. 5a) that clearly exceed the limits of uncertainty.

Fig. 5d also includes the evolution of the squared distance between drifters $B_1$ and $B_5$ that belong to different clusters but nevertheless have overlapping periods of travel time. The fitted power law with an exponent close to one indicates a diffusive regime with linear growth of squared separation. This would be expected for separation distances larger than the typical size of relevant eddies, when uncorrelated velocities imply a constant relative diffusivity (e.g. LaCasce, 2008). According to Fig. 4d the two drifters stay always within or at least in the immediate vicinity of wind farm Global Tech I so that wind farm related
turbulence could possibly explain diffusive behaviour at drifter separations between roughly 3 and 8 km observed in this case.

### 3.1.3 Drifter set C

In experiment C, five drifter pairs were deployed at different locations along a south-north transect west of wind farm Global Tech I (Fig. 6). Unlike the other two experiments, experiment C included periods of rather weak wind conditions (see Fig. 6c). All drift trajectories are characterized by persistent transports to the north-east. Generally, simulations tend to underestimate
the eastward transport components but successfully represent a south-north gradient of the northward drift velocity component.

Squared separations reveal large differences between the five drifter pairs (Fig. 7). Only for one pair ($C_5$, $C_6$) relative dispersion growing exponentially seems a reasonable assumption (Fig. 7c), for all other pairs a less systematic non-monotonic behaviour is observed (the time series for drifter $C_3$ is too short for an assessment). Fitting the exponential growth model for squared distances between drifters $C_5$, $C_6$ much depends on times when drifters have already left the wind farm but may still
feel wind farm related turbulent wakes. The origin of short term decreases of distance during the first day of the drifter journey remains unclear.

Fig. 8 shows the time evolutions of squared distances between drifters selected from different pairs released at different locations. Due to the regular spacing of drifter pair release points, the regrouped drifter pairs fall into classes with initial





**Figure 4. (a)** Observed trajectories $B_1$, $B_2$ and $B_3$. **(b)** Simulated trajectory $B_1$. **(c)** Wind conditions during the observational period. In addition the panel indicates travel times of all trajectories. **(d)** Observed trajectories $B_4$, $B_5$. **(e)** Simulated trajectory $B_5$. Different colours are used for a consistent segmentation of the observational period.



**Figure 5. (a-c)** Time evolutions of pairwise distances between members of the drifter triplet $B_1$, $B_2$ and $B_3$. Dashed lines indicate the fitted exponential growth models annotated in each graph. In the two cases with drifter $B_2$ involved, data in July are ignored as at that time the drifter presumably lost its drogue. **(d)** Distance between drifter pair $B_4$, $B_5$. For comparison, the panel also contains the distance between the two drifters $B_1$ and $B_5$ that belong to different clusters. Colours used for segmentation of the observational period agree with those used in Fig. 4.



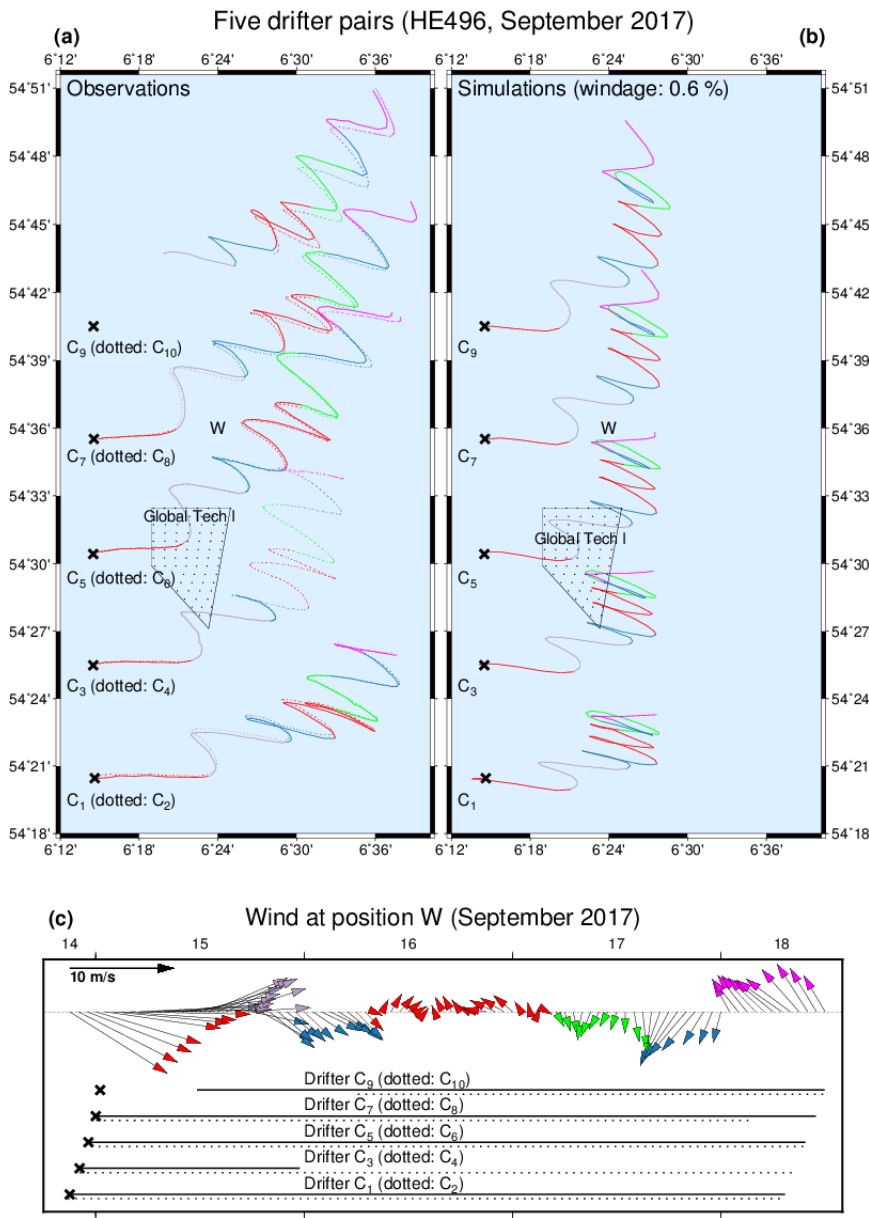

**Figure 6. (a)** Trajectories of five drifter pairs. **(b)** Corresponding simulations. **(c)** Wind conditions during the observational period. Horizontal lines indicate drifter travel times. Different colours are used for a consistent segmentation of the observational period.







**Figure 7.** Panels show for each drifter pair the time dependent squared distance between them. For drifter pair $C_5$, $C_6$ an exponential growth model was fitted, with an e-folding time as indicated in the graph. Colours used for segmentation of the observational period agree with those used in Fig. 6.



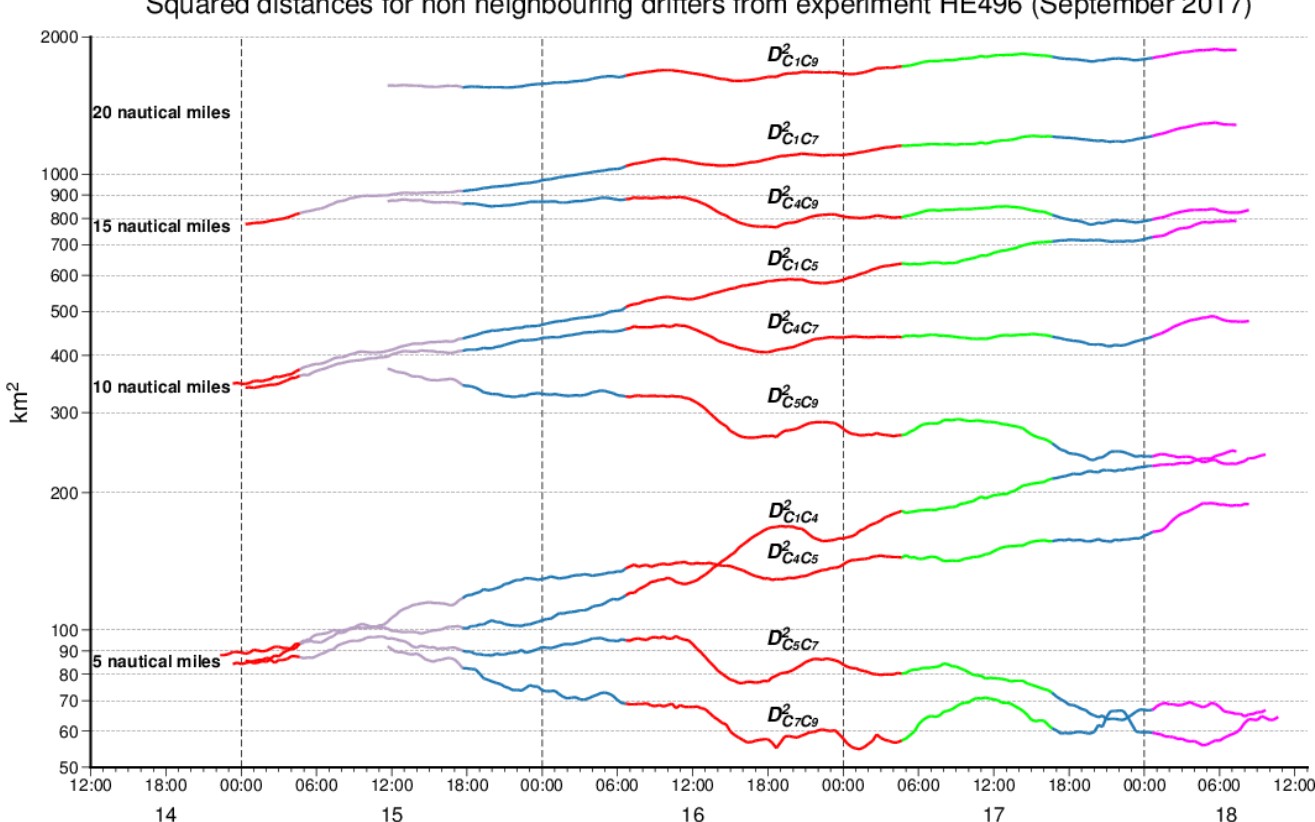

**Figure 8.** Time evolutions of squared distances between drifters not released together (members of different drifter pairs). Colours used for segmentation of the observational period agree with those used in Figs. 6 and 7.

distances of approximately 5, 10, 15 or 20 nautical miles. Even for the same initial separation drifters are found to disperse very differently, trends even differ in sign. Averaging such observations would obviously not provide meaningful insights.

### 3.2   Kinetic energy spectra

Fig. 9 shows a power spectrum of Eulerian velocities observed at research platform FINO3 (see Fig.1) during a two months period . The station is located next to where drifters from drifter set A were deployed. Although the time period underlying Fig. 9 does not overlap with our field experiment, the spectrum nevertheless summarizes the general characteristics of kinetic energy at that location.

The spectrum shows a broadened peak around the frequency of the lunar semidiurnal tide $M_2$ which is the principal tidal constituent in European continental shelves. In addition a clear signal of overtide $M_4$ occurs, higher harmonics are only weakly recognizable. Overtides play a major role for shallow sea tidal variability and are also relevant in the German Bight region (e.g.




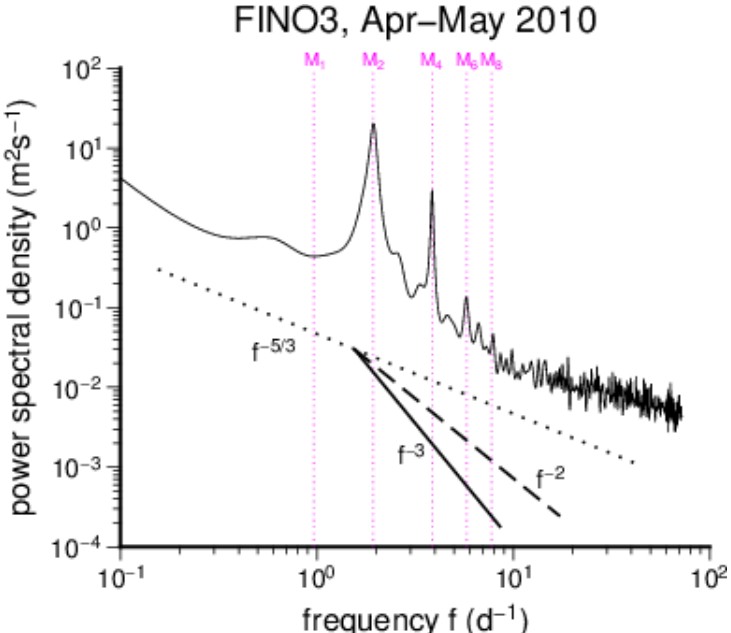

**Figure 9.** Power spectrum of Eulerian velocities observed at research platform FINO3 (see Fig. 1). Magenta coloured lines indicate frequencies of tidal constituents. Auxiliary black lines indicate reference spectral slopes.

Stanev et al., 2014). They are generated by tidal distortion due to non-linear mechanisms of either advection, causing all even harmonics such as $M_4$, or friction, causing odd harmonics such as $M_6$ (Andersen, 1999).

According to Callies and Ferrari (2013) it is important for better understanding of the role of submesoscale turbulence to know how motions represented in the Eulerian spectrum project onto spatial scales. In a first step we compare the Eulerian

energy spectrum (Fig. 9) with its Lagrangian counterparts. Fig. 10 shows Lagrangian velocity spectra analysed from four different drifters. Fig. 10a refers to drifter $A_5$ that is not subject of our study on drifter pairs. Drifter $A_5$ travelled, however, for nearly 49 days (see Callies et al., 2017) so that the length of data recorded compares to the time span underlying the Eulerian spectrum in Fig. 9. In the low frequency range spectral slopes (approximately -5/3) seem similar in the Eulerian and Lagrangian framework, respectively. Note, however, that these low frequency estimates are not very robust considering the limited lengths

of time series. In the high frequency range a theoretical spectrum with slope -2 (Landau and Lifshitz, 1987) approximates the Lagrangian data reasonably well. The Eulerian spectrum flattens out at its high frequency end beyond the tidal modes, reaching a slope of slightly less than -5/3 which would be expected for an inertial energy cascade.

Panels (b)-(d) in Fig. 10 refer to three drifters from our present study. Although these drifters travelled for much shorter times, the spectra found are again at least not in obvious contradiction with an assumed theoretical $f^{-2}$ spectrum. It must

be noted, however, that uncertainties are high and that the spectrum for drifter $A_2$ (not shown), for instance, could also be approximated by $f^{-5/3}$. A finding consistent among all drifters including reference drifter $A_5$ is that the $M_2$ peak in the





**Figure 10.** Power spectra of Lagrangian velocities observed for four selected drifters. Auxiliary black lines indicate reference spectral slopes. Vertical magenta lines indicate frequencies of tidal constituents.

Eulerian spectrum is less dominant or smoother in the Lagrangian spectra. Instead, sharp peaks of overtides up to even $M_8$ are much more pronounced in Lagrangian than in Eulerian spectra.

### 3.3 Velocity increments and structure functions

While single point velocity fluctuations are often close to a Gaussian distribution (e.g. LaCasce, 2005), this is often not true

5   for two-point statistics (e.g. Tsinober, 2001, his Fig. 7.3). Fig. 11 shows the distributions of both longitudinal and transverse





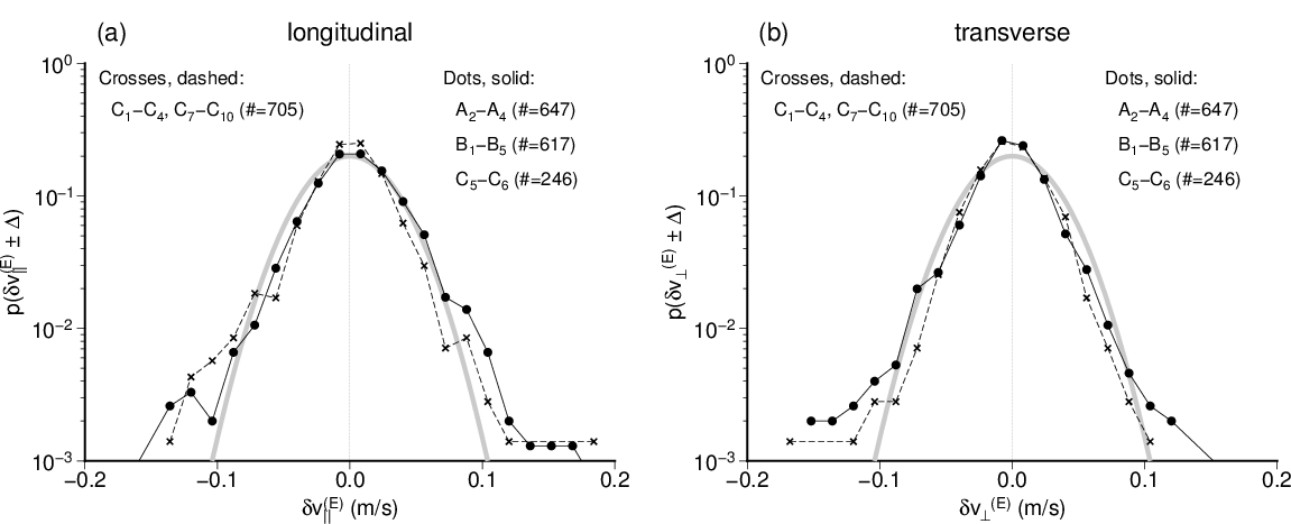

**Figure 11.** Probability distribution functions of **(a)** longitudinal and **(b)** transversal components of Eulerian separation velocities. Velocities grouped in bins of $2\Delta=0.016$ m/s width were evaluated for drifter separations $r < 2500$ m. Solid lines (dots) refer to drifters that were particularly close to wind farms, dashed lines (crosses) to those that presumably did not experience direct wind farm effects. For each group of drifters the number of pairwise samples it contributed is indicated. A reference normal distribution (indicated in grey) was fitted to the longitudinal component (panel (a)) and then just copied to panel (b) as a reference that facilitates a comparison of longitudinal and transverse velocity components.

components of separation velocity. The analysis refers to a subset of data conditioned by drifter separations below 2500 m, which is roughly the maximum distance drifters released as pairs reach within the few days considered. It excludes, however, combinations of drifters deployed at different locations (experiments B and C).

According to Fig. 11 probability distribution functions of longitudinal and transverse Eulerian separation velocities are both

5  nearly Gaussian and not very different from each other. Both graphs in Fig. 11 also distinguish between drifter pairs in close vicinity to wind farms (separating exponentially in time) and others (from drifter set C, separating non-monotonically). However, results for these two groups are very similar, slight differences can possibly be attributed to different weather conditions under which observations were taken. Longer tails of distributions indicate probabilities of fast divergence or convergence being slightly higher than expected for strictly Gaussian distributions. However, distributions in Fig. 11 do not show the pro-

10  nounced exponential tails Poje et al. (2017) analysed from Grand LAgrangian Deployment (GLAD) data in the northern Gulf of Mexico in particular at small separation scales.

The limited number of drifters travelling pairwise can be one motivation for considering Lagrangian velocity increments along single trajectories instead of Eulerian velocity increments between trajectory pairs. Results are shown in Fig. 12. All velocity increments refer to a time delay $\tau = 20$ min, which is the maximum time resolution of our data. Note that sample sizes

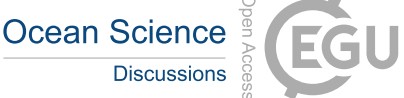



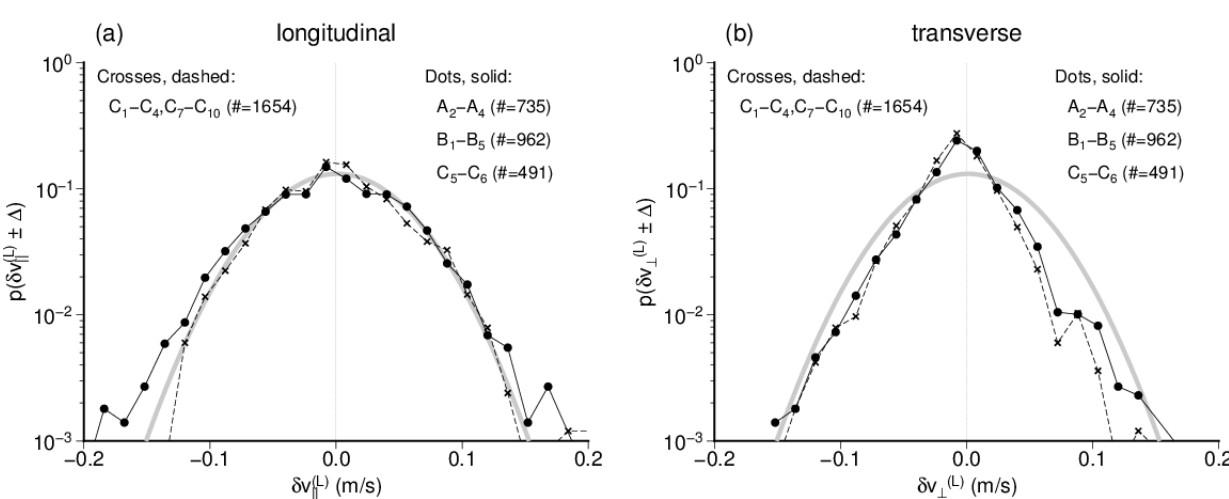

**Figure 12.** Distributions of **(a)** longitudinal and **(b)** transversal components of Lagrangian separation velocities, calculated from binned velocities (bin width $2\Delta$=0.016 m/s) based on 20 min time resolution. Solid lines (dots) refer to those drifters that were particularly close to wind farms, dashed lines (crosses) to those that presumably did not experience direct wind farm effects. A normal distribution fitted to the longitudinal data (indicated in grey) was just copied from panel (a) to panel (b) as a reference that facilitates a comparison of distributions of the two velocity components.

annotated in Fig. 12 are larger than those in Fig. 11 because in the Lagrangian framework data just one drifter is considered as opposed to two in the Eulerian framework.

Like in the Eulerian framework, distributions of Lagrangian longitudinal separation velocities look smooth and nearly normal with, however, slightly enhanced probabilities of large positive or negative values. Distributions obtained from different
sets of drifters are again very similar. By contrast, distributions of transverse velocity components (Fig. 12b) do not just replicate the corresponding longitudinal distribution as they did in the Eulerian framework (Fig. 11b). Instead, the distributions of transverse Lagrangian velocity increments look more triangular (with more exponential wings) than their longitudinal counterparts (Fig. 12a). They also show a preference of negative values indicating counter-clockwise rotation. The latter possibly arises from $M_2$ tidal ellipses which in the German Bight preferably rotate counter-clockwise (Stanev et al., 2014).
Returning to the Eulerian framework, Fig. 13 analyses expected drifter separation velocity as function of spatial distance $r$, considering the second-order structure functions $S_{2,\|}^{(E)}(r)$ and $S_{2,\perp}^{(E)}(r)$ (see Eq. (5)). Like in Figs. 11 and 12 the analysis again distinguishes between two groups of drifters classified in terms of their distances to wind farms. Auxiliary dashed lines indicate the 2/3 slope expected from standard K41 scaling of inertial range turbulence within an either forward (3D) or inverse (2D) energy cascade. An alternative model ($\sim r^2$, dotted lines) is associated with the assumption of a direct enstrophy cascade in
two-dimensional turbulence (see Eq. (8)). Based on the limited amount of data available it is impossible to fit any meaningful power law to the data. Nevertheless, some conclusions suggest themselves.



The most striking feature that occurs for both $S_{2,\parallel}^{(E)}$ and $S_{2,\perp}^{(E)}$ is a plateau like structure in the range of roughly $r = 50 -$ 1000 m. This range falls within the distance between individual turbines of the wind farm. The structure seems most pronounced for the transverse structure function analysed for drifters in close proximity to a wind farm (Fig. 13b). Although some data points suggest a steeper slope for very small distances $r$, the hypothesis of a two-dimensional enstrophy cascade (scaling $\sim r^2$)

cannot be substantiated based on Fig. 13.

For data from experiment C, values of the longitudinal structure function $S_{2,\parallel}^{(E)}(r)$ are too scattered to support the assumption of a plateau (Fig. 13c). On the other hand, for the transverse component $S_{2,\perp}^{(E)}(r)$ even all values in the range of up to 1000 m could be assumed to be on a similar level given the degree of uncertainty indicated in the plot (Fig. 13d). As experiment C includes releases from different locations, Figs. 13c and d cover a larger range of values $r$ than Figs. 13a and b. For large values

of $r$, the transverse structure function $S_{2,\perp}^{(E)}(r)$ in Fig. 13d seems to increase with approximately $r^{2/3}$ as expected for an inverse 2D energy cascade, for instance. Surprisingly, for the longitudinal component (Fig. 13c) this $r$-dependence is missing, values of $S_{2,\parallel}^{(E)}(r)$ tend to remain on a similar level as for smaller distances.

We did not consider Lagrangian counterparts of the Eulerian structure functions shown in Fig. 13. So far a general consensus about possible scaling laws of Lagrangian velocity structure function seems to be lacking (e.g. Biferale et al., 2008; Falkovich

et al., 2012). Another reason is that with increasing values of travel time increment $\tau$ the contributions from tidal currents will start to dominate Lagrangian single-particle velocity differences.

### 3.4   Simulated drifter dispersion

Taking drifter $A_2$ as an example, Fig. 14 shows the evolution of simulation error in terms of squared separation. Surprisingly, again an exponential model fits quite well, even e-folding time 0.64 days resembles those between different drifters. For

comparison, Fig. 14 shows also the simulated spread of a particle cloud, obtained by using a random walk stochastic model superimposed to simulated mean Eulerian currents (Eq. (10)). After a short phase of very quick spreading from a common source point, for a period of approximately one day relative dispersion $D^2$ (Eq. (3)) of the particle cloud develops in a way similar to simulation error. Later on, however, a simulated linear growth of $D^2$ clearly underestimates the increase of simulation error.

Replacing the random-walk by a random-flight stochastic model (Eq. (12)), the period with reasonable rates of spreading can be adjusted by changing the values of Lagrangian decorrelation time $T_L$. Fig. 14 shows example simulations obtained with Lagrangian decorrelation times $T_L$=3 h (analysed by Ohlmann et al., 2012, for instance) or $T_L$=24 h. For larger values of $T_L$ it takes longer until drifters lose memory of their initial turbulent velocities (zero in our example). Therefore initial turbulent velocities are another tuning parameter of the random flight model (together with diffusivity $K$). In the long term, however,

simulated drifter separation will always increase less fast than exponentially.





**Figure 13.** Second order velocity structure functions depending on drifter separation $r$. The range of distance $r$ was subdivided into 25 bins with constant width on a logarithmic scale. Bins populated with less than 10 data point were ignored in the plots. Values are given for longitudinal (left) and transverse (right) components separately. Panels at the top or bottom combine drifters in the vicinity of wind farms **(a, b)** and those travelling more distantly **(c, d)**, respectively. Error bars represent standard deviations estimated by bootstrapping with sample size 1000.





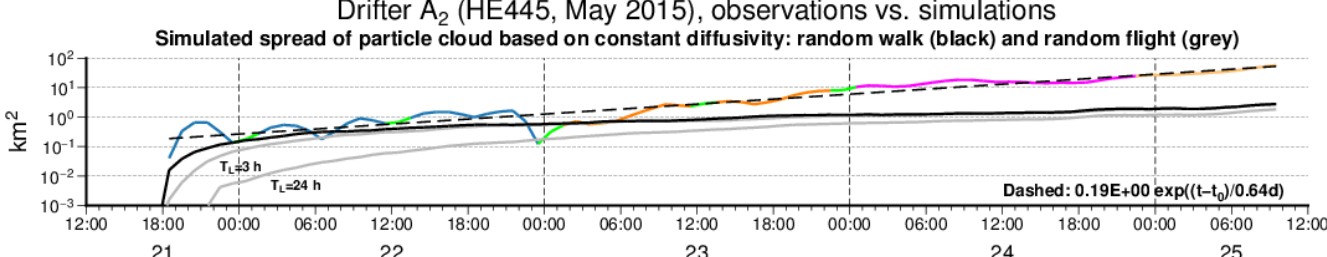

**Figure 14.** Time evolution of squared separation between observed and simulated trajectories of drifter $A_2$. Segmentation using different colours is consistent with that in Fig. 3. The dashed line indicates the fitted exponential growth model annotated in the graph. The solid black line represents relative dispersion $D^2$ obtained from 100 trajectories initialized at the same location and simulated with random walk model Eq. (10). Grey lines show results when using random flight model Eq. (12) with $T_L = 3\,h$ and $T_L = 24\,h$, respectively.

## 4 Discussion

### 4.1 Drifter separations

Coastal currents can be complex and corresponding drifter experiments more site specific than open ocean experiments evaluated by Corrado et al. (2017), for instance. In particular the identification of relevant scales may be affected by regional

bathymetry. Diverging time evolutions within the same 'bundle' in Fig. 8 convincingly illustrate how averaging dispersions of drifter pairs with same initial separation can sometimes be non-informative. Lumpkin and Elipot (2010) discuss some examples of 'atypical' drift trajectories influenced by mesoscale flow features in the North Atlantic and state that at larger scales separation is often non-monotonic.

Pooling roughly 75 drifter pairs deployed with 5-10 m spacing in the Santa Barbara Channel, Ohlmann et al. (2012) observed

circulation to change substantially on a scale of few kilometres. They found exponential growth of mean square pair separation until separation reached a value of approximately 100 m after just 5 hours time. Thereafter alternative models of $D^2$ growing either quadratically or exponentially (e-folding time 0.38 days) both fitted the data reasonably well for the period 12 to 30 hours. Regarding scales in space and time, our data correspond with this latter period. However, in our case exponential models (Figs. 3 and 5) seem acceptable for three days and more, possibly because drifters in experiments A and B stayed

within areas smaller than relevant mesoscale hydrodynamic structures.

All drifter pairs we studied could clearly be classified into those with exponential separation and others with non-monotonic behaviour. Noisy scatter of relative dispersion occurs at times when drifter separation is still below approximately 100 m. At this scale averaging over larger ensembles seems indispensable to achieve a stable statistical characterization, errors in drifter localization may be relevant for the analysis. In the longer term, however, distinction between those pairs that seem

to obey the exponential law and those that do not (for non-monotonic growth also Richardson's power law would not be a meaningful alternative) is surprisingly clear. For eight out of twelve individual drifter pairs assuming that relative dispersion



grows exponentially matched observations well. Except for one instance (e-folding time 0.97 d) all e-folding times fell into the narrow range between 0.45 and 0.56 days. This suggests that the different behaviours we observed might have a physical background.

Many studies find the Rossby radius of deformation to separate exponential growth of pair separation from a Richardson growth regime. Data from the Gulf of Mexico Surface Current Lagrangian Program (SCULP), for instance, provided a large set of 140 drifter pairs (the majority of them being chance pairs) with initial separation below 1 km (LaCasce and Ohlmann, 2003; LaCasce, 2008). Given clear evidence of $D^2$ growing exponentially for about 10 days (e-folding time roughly two days), LaCasce and Ohlmann (2003) suggest injection of enstrophy at the spatial scale of 40-50 km (deformation radius in the Gulf) which then cascades down to smaller scales in agreement with the assumption of non-divergent 2D-turbulence.

Koszalka et al. (2009) analysed exponential growth with an e-folding time of 0.5 days (like in our study) from drifter pairs and triplets deployed within the POLEWARD experiment (2007-2008) conducted in the Nordic Seas. Starting from initial distances < 2 km the phase of exponential separation lasted for just two days up to a final distance of approximately 10 km, in agreement with the size of the local deformation radius. Also Schroeder et al. (2011) analysed e-folding times of 0.5-1 days from drifter clusters deployed in the Liguro-Provençal basin (Mediterranean Sea). These authors found exponential growth lasting for 4-7 days until drifter separation reached a value comparable with the scale of mesoscale circulation patterns (10-20 km in that region).

Also for drifters released near the Brazil Current, Berti et al. (2011) found exponential growth of relative dispersion (e-folding time ~3 days) at scales comparable to the Rossby radius of deformation (~ 30 km). They identified, however, also a second exponential growth regime (e-folding time ~1 day) on a much smaller scale of $\mathcal{O}(1)$ km, assumed to be related to submesoscale flow structures. Studying surface drifter pairs released during the CLIMODE experiment in the Gulf Stream region, Lumpkin and Elipot (2010) found weak evidence for such exponential relative dispersion at scales < 2 km (e-folding time roughly one day). On larger scales up to the Rossby deformation radius (~30 km) drifter separation did clearly not grow exponentially. Lumpkin and Elipot (2010) discuss possible reasons for this discrepancy with other studies, including the use of chance pairs or insufficient temporal spacing of data.

Taken all together, results on relative dispersion at submesoscale are still inconclusive. Uncertainties are high and results from different studies may be conflicting. According to Haza et al. (2008), whether or not an exponential regime can be identified may also depend on the sampling strategy underlying the analysis. Recent comprehensive analyses by Poje et al. (2014, 2017) or Corrado et al. (2017) illustrate the present state of knowledge.

Rich data from the GLAD experiment conducted in Gulf of Mexico from July to October 2012 provides for 300 CODE drifters positions and two-point Lagrangian velocities with high resolution in both space (< 10 m) and time (15 min) (Özgökmen and CARTHE, 2012). From an analysis of these data, Poje et al. (2014) reported evidence that at scales <10 km surface drifter dispersion was driven locally by the effects of eddies comparable in size with drifter separation. In agreement with Richardson's law this implies a shallower spectrum of Eulerian kinetic energy than it would be expected for non-local exponential drifter dispersion we found in our study. Poje et al. (2017) further elaborate on this idea, emphasizing the relevance of ageostrophic submesoscale motions for bypassing the quasi-geostrophic inverse energy cascade.





By contrast, conducting a comprehensive analysis of surface drifter data from the NOAA Global Drifter Program (GDP) Corrado et al. (2017), employing finite-scale Lyapunov exponents (Aurell et al., 1997) to resolve spatial scale dependence, came to the conclusion that exponential growth of drifter pair separation can be found in all parts of the global ocean on spatial scales below the Rossby deformation radius. However, at the sub-mesoscale they found dispersion rates one order

of magnitude larger, corresponding with an e-folding time of roughly 0.5 days for $D^2(t)$. Corrado et al. (2017) suggest the presence of structures in the Eulerian current field that are similar in size to trajectory separation. Existence of two distinct exponential growth regimes could reflect presence of a spectral gap between mesoscale and submesoscale transport regimes (Özgökmen et al., 2012, their Fig. 2).

Oscillatory tidal currents are dominant components of drifter transport in the German Bight (see Fig. 2a, for instance),

similar to wind driven inertial oscillations in the GLAD data (Poje et al., 2014, 2017) which in that region may be difficult to separate from diurnal tidal motions (Gough et al., 2016). However, we found direct manifestation of oscillatory tides being restricted to small short-term dips, color coded (green) in Fig. 3, for instance. According to Fig. 2a these short-term drifter convergences all occurred during periods when tidal currents were oriented towards the south-east, possibly pointing towards bathymetry related effects. A hypothesis to be tested is whether stirring effects by evenly distributed turbines in wind farms

are relevant for injecting tidal energy into the turbulent system. It can plausibly be assumed that a regular stirring process via straining would generate filaments of vorticity expected to be seen in the presence of a 2D enstrophy cascade (Piretto et al., 2016).

Ridderinkhof and Zimmerman (1992) showed that Lagrangian chaos can be the principal mixing process in shallow tidal seas where tides interact with bottom topography ('tidal random walk'). Although the hypothesis of similar chaotic stirring

cannot be substantiated based on our data, it is at least consistent with the observation that exponential growth was absent only for drifters from experiment C that did not travel in close proximity to the wind farms (compare Figs. 6a and 7). Note that experiments A and B were all conducted in the immediate vicinity of wind farms (see Figs. 2 and 4). It should also be noted that wind speeds (and therefore impacts of wind farms on turbulence) were generally lower in experiment C than in experiments A and B (compare Fig. 6c with Figs. 2b and 4c).

For wind farm forcing being non-local (relative dispersion growing exponentially) turbulent energy should be injected at a spatial scale larger than drifter separation. In fact drifter separations stayed below the distance of individual wind turbines (approximately 800 m) for most of the time drifters were tracked. Also the wind farm as a whole might generate relevant hydrodynamic features at a larger scale. An interesting event at the end of the journey of drifters $A_2$, $A_3$ (Fig. 3a) hints at the potential influence of a flow feature at a scale comparable to the already larger drifter separation at that time ($\sim 2$ km). For a

couple of hours beginning at the end of 24 May the distance between the two drifters increased substantially (Fig. 3a, keep in mind the logarithmic scale of the graph). According to Fig. 2a this occurred because for some hours drifter $A_3$ did not share a north-east drift component with drifter $A_2$. An interesting question is whether this reflects a flow feature due to the presence of the wind farm. It is to be kept in mind, however, that a drifter distance of few kilometres is also near the lower bound of possible values of the baroclinic Rossby radius of deformation reported for the North Sea (Becker et al., 1983, 1999; Badin

et al., 2009).





Finally, it is interesting to see that also the discrepancy between the observed trajectory $A_2$ and corresponding simulations (Fig. 14) develops exponentially. The same holds for $A_3$ and $A_4$ (not shown). Also a comparison of Figs. 2a and c reveals that the distance between observed and simulated trajectories of drifter $A_2$ grows at a rate comparable with the growth of distance between drifters $A_2$-$A_3$. In the particular case there is probably little scope left for further improvement of simulations.

## 4.2 Kinetic energy spectra

Some aspects of the Lagrangian velocity spectra in Fig. 10 resemble results that Lin et al. (2017) obtained in their analysis of data from the GLAD experiment in the Gulf of Mexico (Özgökmen and CARTHE, 2012; Poje et al., 2014, 2017). Lin et al. (2017) identified two spectral ranges with different spectral slopes separated by a (in that case diurnal) tidal peak. A $f^{-5/3}$ model approximated the data in the low frequency range, which parallels our finding. For the high frequency range Lin et al. (2017) identified a spectrum with an exponent of about -2.75.

In the Gulf of Mexico study the two spectral ranges were sharply separated at the frequency of a diurnal oscillation. Lin et al. (2017) speculate that the tidal oscillations inject energy which then may cascade towards both smaller and larger scales. In our study we were in the favourable position to have direct measurements of Eulerian spectra (Fig. 9) that could be indicative of such cascade dynamics. On the other hand, the German Bight tidal regime is more complex than that in the Gulf of Mexico. According to Fig. 9 it seems that three rather than just two spectral ranges should be distinguished. In an intermediate frequency range between about 2 and 8 $\mathrm{d}^{-1}$ a spectral slope of approximately $f^{-2}$ occurs with sharp peaks related to the principal tidal constituent $M_2$ and at least overtide $M_4$. A spectral slope of -3 would be in agreement (using Taylor's frozen turbulence transformation $k \sim f/u$, where $u$ denotes mean velocity) with the assumption of a direct enstrophy cascade in 2D turbulence (see Eq. (6)). Here, however, tidal energy input can obviously not be described as being local in the frequency domain, overtides injecting energy at frequencies higher than $M_2$ may possibly reduce the spectral slope. Based on numerical simulations for a two month period without extreme atmospheric conditions, Stanev et al. (2014) found that an area of major $M_4$ amplitudes off the North Frisian Wadden Sea originated from reflection and refraction of the Kelvin wave underlying the North Sea $M_2$ tide. Large $M_6$ velocity components were found to occur in estuaries and tidal channels with strong velocities and high friction. Such energy transfers between tidal constituents are clearly not a matter of pure turbulence expected to follow classical scaling laws.

For high frequencies beyond the range of tidal signals a Eulerian power spectrum of even less than -5/3 is observed in Fig. 9. With Taylor's frozen turbulence assumption, a -5/3 slope would reproduce Kolmogorov's law (Eq. (6)). Although this law can be found for very different systems (Tsinober, 2001, Section 7.3.4), it is also theoretically consistent with either a direct energy cascade in fully developed 3D turbulence or an inverse energy cascade in 2D turbulence. A -5/3 slope in Eulerian spectra would also be consistent with the fact that slopes in the Lagrangian spectra (Fig. 10) seem to be close to -2, predicted by the Kolmogorov-Landau theory (Landau and Lifshitz, 1987) and confirmed experimentally for fully developed 3D turbulence (e.g. Mordant et al., 2001, 2003).

The low frequency part of the Eulerian spectrum in Fig. 9 is poorly underpinned by data and must be interpreted with due care. However, surprisingly the Lagrangian spectrum (Fig. 10a) seems to replicate a -5/3 slope of the Eulerian spectrum. A





-2 slope expected theoretically in a Lagrangian framework derives from dimensional arguments, exponents in the Eulerian and Lagrangian framework differ because only in the Eulerian context the spectrum is assumed to depend on a mean velocity (Landau and Lifshitz, 1987, p. 135). However, a -5/3 Lagrangian spectrum at low frequencies was also found by Lin et al. (2017) in their Gulf of Mexico study. As a possible problem these authors mention the presence of tidal movements which

according to Beron-Vera and LaCasce (2016) can cause conflicting results between different types of analyses.

Middleton (1985) addresses the general question how Eulerian spectra translate into their Lagrangian counterparts. Elaborating on an original idea of Corrsin (1959), Middleton (1985) found that spectra observed in an Eulerian and Lagrangian framework, respectively, should agree when velocity changes depend more on local variations than on advective processes (see also LaCasce (2008) for a summary of the concept). This situation might prevail with the scales involved in tidal movements.

Off the coast, spatial scales over which tidal currents change are larger than the tidal excursions of individual water bodies, which implies a minor role of advective processes.

### 4.3 Velocity increments and structure functions

A problem we are faced with is that velocity structure functions in Fig. 13 do not show the scaling with $r^2$ expected for non-local (i.e. exponential) relative dispersion we observed for most of our drifter pairs (see Eqs. (7) and (8)). Fig. 13 suggests

a fast increase of $S_2^{(E)}$ just for very small distances before the structure function levels off towards a plateau-like behaviour. Although our data are insufficient for fitting statistical models, for parts of the spectra shallower slopes $\propto r^{2/3}$ seem more consistent with observations. A similar situation has also been reported in other studies based on larger sets of data (Beron-Vera and LaCasce, 2016), even when more sophisticated distance based measures like the finite-scale Lyapunov exponent (FSLE, see Aurell et al., 1997) were employed (Sansón et al., 2017). Beron-Vera and LaCasce (2016) understand their study as

a warning not to deduce kinetic energy spectra from measurements of relative dispersion. To explain the seeming discrepancy, they proposed two different effects. First, values of distance $r$ between drifters deployed pairwise do often not cover the whole range up to the mesoscale structure where energy for non-local forcing is injected. In our case one might argue that this range is covered by combinations of drifters from different pairs in experiment C, which provide the instances of large $r$ values in Figs. 13c and d. However, these values are not indicative of structure functions growing faster than $r^{2/3}$.

A second explanation Beron-Vera and LaCasce (2016) propose is that $S_2^{(E)}(r)$ values for small values of $r$ are larger due to the effects of (in their experiment) regular inertial oscillations. The mechanism proposed is that for constant angular velocity the size of an inertial loop should correlate with drifter velocity, which may give rise to a correlation between drifter separation and separation velocity while mean dispersion after closed cycles remains unaffected. However, the strong externally forced tidal oscillations in our experiments vary smoothly in space and neighbouring drifters are supposed to experience similar

movements. Also the spatial scale of tidal waves seems clearly larger than the separations of mostly less than 1 km reached by most of our drifter pairs within the drift period of 3-4 days. Given the large tidal excursions (see Fig. 2a, for instance), in our case tidal movements cannot be seen as small scale disturbances overlaid to large scale movements. The situation seems different in experiment C (see Fig. 6a). However, for analysing such large scale homogeneous movements our time series of 3-4 days lengths are too short.





For isotropic turbulence the following relationship should relate the longitudinal and transverse second-order structure functions to each other (Kraichnan, 1966; Monin and Yaglom, 1975):

$$S_{2,\perp}^{(E)}(r) = \left(1 + \frac{r}{2}\frac{d}{dr}\right) S_{2,\parallel}^{(E)}(r) \tag{13}$$

Kramer et al. (2011) propose verifying Eq. (13) for checking the assumption of homogeneity and isotropy. Obviously our data
are too noisy for following this approach, which according to Babiano et al. (1985) would not be fully conclusive anyway. It must also be noticed that for large drifter separations (up to 40 km in experiment C), systematic spatial patterns of the tidal regime may dominate the analysis in Fig. 13. It can reasonably be assumed that rotational components of tidal currents preferably impact the transverse components of velocity increments (Lévêque and Naso, 2014). Resulting dependences might happen to resemble what one would expect from statistical analyses.

## 4.4 Simulated drifter dispersion

Fig. 14 exemplifies simulation error growth for drifter $A_2$. Interestingly, also simulation errors for drifters C grow exponentially with similar e-folding times (not shown), notwithstanding the irregular behaviour of observed relative dispersion (Fig. 7). Comparing Figs. 6a and b reveals a (possibly location dependent) lack of eastward transport in simulations, which means that the observed and simulated drifters, respectively, persistently experience different large scale background currents. This is
reminiscent of exponential growth rates that occur when distances between drifters are stretched by eddies much larger in size than the distance between two drifters considered.

Simulation errors exceed simulated random spread of drifters. Simulations that employ an either zeroth-order (Eq. (10)) or first-order (Eq. (12)) stochastic model both underestimate drifter separation after more than about two days, while overestimating drifter separation in the very first hours after drifter deployment (Fig. 14). A clear distinction between processes resolved
by the hydrodynamic model and sub-grid scale processes to be parametrized may be missing. Instead of assuming constant diffusivity, Ohlmann et al. (2012) used turbulent velocity standard deviations $\sigma$ ranging between 0.7 and 5.1 cm/s depending on separation scales between 5 m and 2 km. With $T_L$ = 3 h, this corresponds with values of diffusivity $K = \sigma^2 T_L$ approximately ranging between 0.5 and 28 $\mathrm{m}^2/\mathrm{s}$. The lower bound of these values corresponds with the magnitude of the value obtained from Eq. (11) with grid resolution 900 m used in our simulations.

## 5 Conclusions

The most striking finding from the analysis of eleven trajectory pairs released in the German Bight was that these could very clearly be grouped into eight pairs showing exponential increase of drifter separation and three pairs distances of which changed in a non-monotonic way. One pair travelled too short for a clear assessment. For seven out of the eight pairs a fitted e-folding time for squared separation was found to be approximately half a day, for the eighth drifter pair the e-folding time was about
twice as large.





In light of this classification we refrained from a statistical analysis considering all drifter pairs as independent realizations of the same generic behaviour. Reasons for the differences we found can just be speculated. One hypothesis is that effects of wind farms manifest themselves in drift behaviour (exponential separation rates). Even when this hypothesis cannot really be substantiated based on the limited amount of observations, it is nevertheless consistent with the observation that none of the

three pairs with non-monotonic growth travelled within a wind farm or in close neighbourhood on its lee side.

Non-monotonic drifter separation could possibly be indicative of drifters getting trapped by coherent structures. Elhmaïdi et al. (1993) decomposed simulations of 2D turbulence with regard to either deformation or rotation dominating the local hydrodynamic structure and proposed the use of conditional averages. Shelf sea conditions depending on irregular coastal geometry and bathymetry must be expected to manifest themselves in characteristic hydrodynamic structures at specific spatial

scales. Indeed, already on the scale of say 5-10 nautical miles we found drifter behaviour to reflect influence of mesoscale flow patterns (see Fig. 8). Under these conditions a statistical analysis of evolving drifter separation on scales beyond few nautical miles is a questionable enterprise. A threshold of scale separation can possibly be derived from a plateau-like structure only hinted at, however, in the Eulerian second order structure function. The scale separation at $\mathcal{O}(1)$ km overlaps with distances between individual turbines in wind farms but is also not far from the magnitude of the internal radius of deformation, which

in the German Bight is estimated to be few kilometres.

Important flow characteristics in the German Bight are strong tidal currents. In the Eulerian kinetic energy spectrum we found peaks of tidal constituents embedded in a spectral range with an approximately -2 slope. Interestingly, in the Lagrangian spectra derived from drifter movements, even peaks related to higher-order overtides $M_6$ and $M_8$ were well defined. Energy injected at different frequencies and nonlinear transformation of energy between different tidal constituents, however, obviously

goes beyond the classic concept of turbulent energy cascading across an inertial spectral range.

Definitely a more systematic field study would have to be designed to disentangle the aforementioned different effects and to possibly identify effects of wind farms on turbulent mixing in the German Bight. The preliminary results of the present analysis could be helpful in designing such field experiment.

*Data availability.* The raw data sets A (HE445), B (HE490) and C (HE496) are freely available from Carrasco and Horstmann (2017) and

Carrasco et al. (2017a, b).

*Author contributions.* JF, JH and RC collected the field data, RC was in charge for data management including quality control and documentation. MQ performed the spectral analyses. UC provided numerical drift simulations and prepared the manuscript with contributions from the four co-authors.

*Competing interests.* The authors declare that they have no conflict of interest.



*Acknowledgements.* The three RV Heincke research cruises were supported by grant numbers AWI_HE445_00 , AWI_HE490_00 and AWI_HE331_00. Drifter simulations were based on BSHcmod currents provided by the Federal Maritime and Hydrographic Agency (BSH). The 10 m wind data used are from the operational forecasting system of the Deutscher Wetterdienst (DWD). We thank BMWi (Bundesministerium für Wirtschaft und Energie) and the PTJ (Projekttraeger Juelich, Project Executing Organization) for making avail-
5   able Eulerian currents observed at FINO3. All graphs were produced using the Generic Mapping Tools software (GMT) available from www.soest.hawaii.edu/gmt/. Ulrike Kleeberg assisted in preparing Fig. 1.





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





**Table 1.** Drifters considered in this study

| Label | Start | | | End | | | Length | Dist | $\Delta T$ |
|---|---|---|---|---|---|---|---|---|---|
| | Time (UTC) | °E | °N | Time (UTC) | °E | °N | [km] | [km] | [days] |
| **HE445 (May 2015):** | | | | | | | | | |
| $A_2$ | 21 May (17:13) | 7.1484 | 55.0752 | 25 May (09:47) | 7.3080 | 55.1360 | 87.4 | 12.2 | 3.7 |
| $A_3$ | 21 May (17:13) | 7.1480 | 55.0750 | 25 May (09:59) | 7.2526 | 55.1160 | 85.7 | 8.1 | 3.7 |
| $A_4$ | 21 May (17:36) | 7.1426 | 55.0786 | 24 May (15:00) | 7.2960 | 55.0626 | 66.6 | 10.0 | 2.9 |
| — | | | | | | | | | |
| $A_5$ | 27 May (09:49) | 5.9126 | 54.3752 | 15 Jul (01:28) | 8.4680 | 55.1232 | 1264.0 | 184.4 | 48.7 |
| **HE490 (June/July 2017):** | | | | | | | | | |
| $B_1$ | 29 Jun (08:09) | 6.2560 | 54.5214 | 3 Jul (05:45) | 6.5864 | 54.4770 | 95.5 | 22.0 | 3.9 |
| $B_2$ | 29 Jun (08:04) | 6.2576 | 54.5212 | 1 Jul (06:10) | 6.4850 | 54.5070 | 49.9 | 14.8 | 1.9 |
| $B_3$ | 29 Jun (08:05) | 6.2574 | 54.5212 | 2 Jul (05:21) | 6.5406 | 54.4918 | 74.1 | 18.6 | 2.9 |
| — | | | | | | | | | |
| $B_4$ | 29 Jun (13:25) | 6.3336 | 54.5214 | 1 Jul (11:14) | 6.3422 | 54.5232 | 46.6 | 0.6 | 1.9 |
| $B_5$ | 29 Jun (13:20) | 6.3322 | 54.5212 | 2 Jul (10:40) | 6.3882 | 54.5272 | 71.4 | 3.7 | 2.9 |
| **HE496 (September 2017):** | | | | | | | | | |
| $C_1$ | 14 Sep (21:01) | 6.2432 | 54.3408 | 18 Sep (07:23) | 6.6272 | 54.4320 | 86.2 | 26.9 | 3.4 |
| $C_2$ | 14 Sep (20:49) | 6.2442 | 54.3412 | 18 Sep (07:12) | 6.6222 | 54.4340 | 86.4 | 26.6 | 3.4 |
| — | | | | | | | | | |
| $C_3$ | 14 Sep (22:09) | 6.2416 | 54.4250 | 15 Sep (23:30) | 6.4238 | 54.4762 | 25.1 | 13.1 | 1.1 |
| $C_4$ | 14 Sep (22:11) | 6.2422 | 54.4250 | 18 Sep (08:32) | 6.5596 | 54.5626 | 83.1 | 25.6 | 3.4 |
| — | | | | | | | | | |
| $C_5$ | 14 Sep (23:10) | 6.2450 | 54.5078 | 18 Sep (09:44) | 6.6208 | 54.6854 | 80.8 | 31.3 | 3.4 |
| $C_6$ | 14 Sep (23:30) | 6.2472 | 54.5082 | 18 Sep (09:53) | 6.6318 | 54.6820 | 80.7 | 31.5 | 3.4 |
| — | | | | | | | | | |
| $C_7$ | 15 Sep (00:01) | 6.2482 | 54.5920 | 18 Sep (10:56) | 6.6446 | 54.7668 | 76.6 | 32.1 | 3.5 |
| $C_8$ | 15 Sep (00:06) | 6.2480 | 54.5920 | 18 Sep (03:32) | 6.5618 | 54.7590 | 69.0 | 27.5 | 3.1 |
| — | | | | | | | | | |
| $C_9$ | 15 Sep (02:22) | 6.2854 | 54.6766 | 18 Sep (11:56) | 6.5972 | 54.8482 | 67.5 | 27.7 | 3.4 |
| $C_{10}$ | 15 Sep (16:50) | 6.4134 | 54.7180 | 18 Sep (12:04) | 6.5994 | 54.8512 | 46.5 | 16.0 | 2.2 |

Drifters released as pairs or triplets during three different field experiments in the German Bight. Initial and final locations were defined according to the list of locations communicated via the satellite communication network. Type: Drifter type used. Length: Sum of the lengths of linear segments connecting observed drifter locations. Dist: Linear distance between the first and the last drifter location observed. $\Delta T$: Days between the first and the last observation. Single drifter $A_5$ is not a subject of the present study but due to its long-lasting journey used as a reference in Fig. 10a.