# Peer review of "Submesoscale dispersion of surface drifters in a coastal sea near offshore wind farms"

_Ocean Science, 2018_

## Referee Comment (RC1) · Anonymous Referee #1 · 26 Dec 2018

This is a very nice paper on dispersion characteristics in German Bight. What makes this paper quite special is the existence of offshore wind farms (OWFs) in the region, even though these effects do not really show up in the results. Also, the authors did an excellent job in executing the paper as well discussing at length with respect to all previous work (maybe with the exception of some, as suggested below). Some of this discussion is fueled by the inconclusive nature of the results, which seem to be mainly due to the small number of drifters; something that could be improved in the future. Nevertheless, overall it is a great, careful study and I recommend acceptance subject to possible modification as per my minor comments below:

* page 3: In a variation of the nice literature review laid out by the authors, the following paper (on the basis of 300 drifters) says that local and non-local anti-dispersion disper-

[Figure]

sion can be imbedded in each other; namely, as the larger scale tracer cloud grows in size, parts of it get concentrated by surface convergence, or ageostrophic motions:

D'Asaro et al., 2018: Ocean convergence and dispersion of flotsam. PNAS, https://doi.org/10.1073/pnas.1718453115.

It is a bit more complicated but perhaps more accurate depiction of what could be going on in the ocean. Especially considering that wakes from OWFs here, maybe it is applicable.

* page 4, paragraph 20: regarding how GPS errors reflect to some dispersion metrics, the following paper has some analysis:

Haza et al., 2014: How does drifter position uncertainty affect ocean dispersion estimates? J. Atmos. Ocean Tech., 31, 2809-2828.

* page 8, paragraph 20: how was beta=0.006 determined? Is it based particularly for these types of drifters? I am asking because typical wind drift is about 3%:

Bye, J.A.T., 1967: The wave-drift current. J. Mar. Res. 25, 95-102. Bye, J.A.T., 1988: The coupling of wave drift and wind velocity profiles. J. Mar. Res. 46, 457-472. Wu, J., 1975: Wind-induced drift currents. J. Fluid Mech., 68, 49-70. Wu, J., 1983: Sea-Surface Drift Currents Induced by Wind and Waves. J. Phys. Oceanogr. 13, 1441-1451.

* page 9, line -2: I do not quite understand this conclusion: is the large amplitude sinusoidal behavior in the trajectories governed by wind or tidal cycle? If wind only, is there no influence of the tides there (I am not familiar with the area)?

* If wind is very important, the authors should explain a bit more whether wind effect is happening due to coefficient beta (which is smaller than I expected) or the model BSHchmod. I am asking this because no ocean model I have seen is very good in simulating current/wind/wave effect in the upper 0.5 m of the water column.

* page 16, figure 6: very impressive agreement between real and modeled trajectories! Can the authors comment why the agreement is so good? Is it the wind, or lack of coherent structures in the ocean (which usually tend to lead to chaos), or..?

* page 28: Veron-Bera and LaCasce (2016) filter at inertial time scales, which coincide with the temporal range of submesoscale. They could be throwing the baby with the bathwater.

---

## Referee Comment (RC2) · Anonymous Referee #2 · 5 Feb 2019

In this paper, an investigation of the properties of relative dispersion, structure functions and spectra is presented, from drifters released in the German Bight. The paper is written in a rather clear and competent way, but the results are in my opinion insufficiently robust and inconclusive.

I think the paper is not publishable in its present form, and it should go through a major revision or a resubmission.

MAIN COMMENT

The data set is relatively small (a total of 19 drifter pairs), and the authors choose to present dispersion properties for each pair independently, attempting to discuss their individual characteristics and statistics. They justify this approach in terms of coastal

inhomogeneity which would prevent a global statistical approach. This hypothesis, though, is not sufficiently substantiated by the data as discussed in the following, and the end result is that the statistics of each pair (with duration of 1-4 days) is too poor to reach robust conclusions.

My suggestion is the following. I think that the authors could indeed start with a description of the individual launches, in terms of geographical positions and wind and tidal forcing, without though going in the details of the individual dispersion plots and fits. After the general presentation, I think the authors should present some clear working hypotheses on parameters that could influence the statistics, that will then be consistently tested throughout the paper. The parameters could be related to topography, forcing or distance from offshore wind farms (OWF). These hypotheses will be tested though conditional statistics, using selected sub ensemble of data. Given the small number of data, the conditional sub ensembles should be as broad as possible, based on the chosen parameter.

The results from these conditional statistics will then be compared with the total statistics obtained from all the pairs, in order to verify whether or not significant differences emerge.

This will provide a logical structure to the paper, and a setting that will allow testing working hypothesis. It might be that the data set is too small and the errors are too big to actually differentiate between conditional statistics, but at least this will be shown in a quantitative way. In the present version of the paper, the authors actually take a similar approach for the discussion of the spectra and structure functions, but the hypotheses are not presented in a clear fashion and are not consistent throughout the paper.

DETAILED COMMENTS

Section 1 Lines 1-5. There are a number of recent papers that investigate "local" initial conditions (e.g. Ohlman et al, 2017; Berta et al., 2016; Poje et al., 2014)

Line 20 Please expand on the mechanisms through which OWF are expected to impact on surface dispersion

Section 2

Lines 10-20. Please discuss expected slippage errors of the MDO3 drifters. Have they been quantitatively tested? and compared with other types of drifters such as the classic CODE? Please provide references

Table 1. It should be improved or complemented by an other table. Initial distances between pairs and distances from OWFs should be included

Also in the text, in Section 2 and 3, please be more quantitative, avoid mentioning that pair are "close" or far, and refer to the i.c. in Table 1

Section 2.4. Please specify model initial distances between pairs and comment on the fact that given a model resolution of 900 m, local structures beyond 2-4 km are not correctly resolved.

Fig. 1. It should be improved, showing the deployment design and the topography

Section 3.

Fig.3 5,7 and related text. The exponential fit seems very arbitrary to me. Were other fits tested as well? The initial distances from which the fit start should be mentioned. Please discuss errors and confidence limits. In order to compare results, the initial distance should be comparable. See also the point on model pairs above. In general, please see General Comment above.

Section 3.2. The computed spectra are in time, while the general discussion in 2.2 is in terms of wavenumbers. Please discuss the hypotheses used to link the two types of spectra. The drifter spectra (except for one case) are obtained from time series of 1-3 days. Can they effectively resolve tidal frequency, even using MMT? Please discuss errors and confidence limits

[Figure]

Section 3.3. What do the authors mean by "Eulerian and Lagrangian" separation?

Section 3.4. What are the initial distances of the model pairs? Given the model resolution, the dynamics is not expected to be local beyond 2-4 km, so that the exponential behavior is simply a consequence of the setting..

———————————————————

---

## Author Comment (AC1) · 5 Mar 2019

The comment was uploaded in the form of a supplement:
https://www.ocean-sci-discuss.net/os-2018-118/os-2018-118-AC1-supplement.pdf

---

## Author Response (AR1)

**Reply to Referee #1:**

We greatly thank the referee for the effort he applied on the review, for helpful suggestions and the provision of additional relevant references.

In the following, the referee's comments are shown in blue.

This is a very nice paper on dispersion characteristics in German Bight. What makes this paper quite special is the existence of offshore wind farms (OWFs) in the region, even though these effects do not really show up in the results. Also, the authors did an excellent job in executing the paper as well discussing at length with respect to all previous work (maybe with the exception of some, as suggested below). Some of this discussion is fueled by the inconclusive nature of the results, which seem to be mainly due to the small number of drifters; something that could be improved in the future. Nevertheless, overall it is a great, careful study and I recommend acceptance subject to possible modification as per my minor comments below:

- page 3: In a variation of the nice literature review laid out by the authors, the following paper (on the basis of 300 drifters) says that local and non-local anti-dispersion dispersion can be imbedded in each other; namely, as the larger scale tracer cloud grows in size, parts of it get concentrated by surface convergence, or ageostrophic motions:
  ➢ D'Asaro et al., 2018: Ocean convergence and dispersion of flotsam. PNAS, https://doi.org/10.1073/pnas.1718453115.
  It is a bit more complicated but perhaps more accurate depiction of what could be going on in the ocean. Especially considering that wakes from OWFs here, maybe it is applicable.
  Indeed this is a very relevant reference, thank you for the advice! In the revised manuscript we refer to this study in both the introduction and the discussion section.
- page 4, paragraph 20: regarding how GPS errors reflect to some dispersion metrics, the following paper has some analysis:
  ➢ Haza et al., 2014: How does drifter position uncertainty affect ocean dispersion estimates? J. Atmos. Ocean Tech., 31, 2809-2828.
  This study has been added as a reference in Section 2.1.
- page 8, paragraph 20: how was beta=0.006 determined? Is it based particularly for these types of drifters? I am asking because typical wind drift is about 3%:
  ➢ Bye, J.A.T., 1967: The wave-drift current. J. Mar. Res. 25, 95-102.
  ➢ Bye, J.A.T., 1988: The coupling of wave drift and wind velocity profiles. J. Mar. Res. 46, 457-472.
  ➢ Wu, J., 1975: Wind-induced drift currents. J. Fluid Mech., 68, 49-70.
  ➢ Wu, J.,1983: Sea-Surface Drift Currents Induced by Wind and Waves. J. Phys. Oceanogr. 13, 1441-1451.
  The value of beta was specified in Callies et al. (2017) considering drifters of the same type. This reference is given in the manuscript. As Callies et al. (2017) dealt with the same type of drifters, one could say that beta was optimized for this drifter type. However, less perfect behaviour would imply a larger rather than a smaller value of beta. Callies et al. (2017) found that assuming a drag of 0.6% of the 10 m wind corresponded with using 50% of surface Stokes drift, which is in reasonable agreement with the fast decrease of Stokes drift with depth and the fact that drifters are supposed to represent currents in a 1m surface layer.
- page 9, line -2: I do not quite understand this conclusion: is the large amplitude sinusoidal behavior in the trajectories governed by wind or tidal cycle? If wind only, is there no influence of the tides there (I am not familiar with the area)?

It seems a typing error has crept in and the referee refers to the discussion in the first paragraph in Section 3.1.1 (lines ~20), where Fig2a is described.

Tidal currents are very important in this region and the large amplitude sinusoidal oscillations (oriented along a southeast-northwest direction) are caused by these tides indeed. However, superimposed to this regular oscillation, wind-driven residual currents move this regular pattern of tidal movements back and forth along a roughly southwest-northeast oriented direction (during the period studied). A reversal of this shift by residual currents occurs, for instance, at the end of the period that is colour coded in blue. We revised the description in the following form: "*After winds veered to blow from the north-west, residual transports reverse their direction and the tide induced pattern of oscillatory drifter movements is shifted back towards the* OWF *area*".

- If wind is very important, the authors should explain a bit more whether wind effect is happening due to coefficient beta (which is smaller than I expected) or the model BSHchmod. I am asking this because no ocean model I have seen is very good in simulating current/wind/wave effect in the upper 0.5 m of the water column.

The reversal of residual currents after winds veered is an effect that does not depend on $\beta$ but is an effect already contained in the hydrodynamic fields from BSHcmod our drift simulations are based on. The following sentence has been added: "*Note that this reversal does not depend on the choice of $\beta$ parametrizing wind drag in Eq. (9) but is already represented by the Eulerian surface currents $v_E$*".

- page 16, figure 6: very impressive agreement between real and modeled trajectories! Can the authors comment why the agreement is so good? Is it the wind, or lack of coherent structures in the ocean (which usually tend to lead to chaos), or..?

Looking at the set of observed trajectories it seems that they all follow a rather homogeneous mesoscale flow pattern. Therefore the relevant flows can be well resolved by the model. The situation might change under other wind conditions. Note that during this experiment winds were relatively weak, possibly giving rise to less chaotic flow patterns. But it is hard to provide a reliable answer without further experiments in this area.

However, a recent study that was just published seems to at least not contradict our estimate of beta. In the manuscript we added the following comment to the paragraph following Eq. (9): "*This value estimated by Callies et al. (2017) seems largely consistent with findings of a more recent experimental study by Meyerjürgens et al. (2019). From seven drifters tracked in the German Bight they estimated a wind slip of 0.27 % and a total wind induced drifter motion of 1 % of 10~m winds.*"

- page 28: Veron-Bera and LaCasce (2016) filter at inertial time scales, which coincide with the temporal range of submesoscale. They could be throwing the baby with the bathwater.

At least their argument does not seem applicable for the data we are dealing with.

**Reply to Referee #2:**

We greatly thank the referee for the effort he applied on his review and for his helpful comments.

In the following, the referee's comments are shown in blue.

In this paper, an investigation of the properties of relative dispersion, structure functions and spectra is presented, from drifters released in the German Bight. The paper is written in a rather clear and competent way, but the results are in my opinion insufficiently robust and inconclusive.

I think the paper is not publishable in its present form, and it should go through a major revision or a resubmission.

MAIN COMMENT

The data set is relatively small (a total of 19 drifter pairs), and the authors choose to present dispersion properties for each pair independently, attempting to discuss their individual characteristics and statistics. They justify this approach in terms of coastal inhomogeneity which would prevent a global statistical approach. This hypothesis, though, is not sufficiently substantiated by the data as discussed in the following, and the end result is that the statistics of each pair (with duration of 1-4 days) is too poor to reach robust conclusions.

We fully agree (and state that in the paper) that, due to the low number of drifters, our findings are not robust in a statistical sense. Fig. 8 shows very clearly how differently drifter pairs with relatively large initial separation (> 9 km) behave. Also for smaller distances (< 1 km) a comparison of Figs. 5 and 7, for instance, suggests that averaging over different drifters would not generate useful information. We agree with Referee #1 that only future experiments could improve the situation. For the time being, we believe that the best that can be done is to summarize all (admittedly weak) indications available.

My suggestion is the following. I think that the authors could indeed start with a description of the individual launches, in terms of geographical positions and wind and tidal forcing, without though going in the details of the individual dispersion plots and fits. After the general presentation, I think the authors should present some clear working hypotheses on parameters that could influence the statistics, that will then be consistently tested throughout the paper. The parameters could be related to topography, forcing or distance from offshore wind farms (OWF). These hypotheses will be tested though conditional statistics, using selected sub ensemble of data. Given the small number of data, the conditional sub ensembles should be as broad as possible, based on the chosen parameter.

The results from these conditional statistics will then be compared with the total statistics obtained from all the pairs, in order to verify whether or not significant differences emerge.

The referee asks for a formalized statistical analysis, testing well-defined hypotheses. We agree that a number of different parameters could influence drifter behaviour. Forcing (weather conditions) undoubtedly is among these important factors. Our analysis combines three experiments at different times. We do not see, however, how different weather conditions at these times could be formally described or characterized. Weather conditions cannot be characterised in terms of just one parameter. During experiment

HE496 wind speeds tended to be smaller than during the other two experiments. Does that already mean that environmental conditions during HE445 and HE490 fall into one class (regarding weather) while conditions during HE496 establish a second class? Experiment HE496 also happens to be the experiment in which drifters travelled at larger distances from the wind farm. How could the impacts of these two factors be separated from each other?

We are afraid that formalizing the study in terms of conditional statistics would generate a substantial formal overhead without promising a clear benefit. Note, however, that in Figs. 11, 12 and 13 we already did some conditioning, showing distributions and structure functions for different groups of drifters, roughly defined in terms of distance from wind farms. This grouping is necessarily qualitative, considering also the fact that these distances change when drifters move. It is in HE496 (drifter set C, Fig. 6) that larger distances from the wind farm occur.

We appreciate the referee's intention to improve the common thread of the discussion. To clarify the general structure of the analysis we included the following introductory paragraph at the very beginning of Section 3 (Results): "*Section 3.1 presents details of all drift trajectories analysed in this study. Plots show how drifters are located relative to wind farms and which winds they are exposed to. In Section 3.2 kinetic energy spectra are studied to assess the possible relevance of tidal movements as a source of turbulent energy. Section 3.3 then presents probabilities of separation velocities and velocity structure functions. To check the hypothesis that drifter separation might be influenced by wind farm related turbulence, these functions are shown for different groups of drifters, separating in particular those drifters that are far enough to presumably not experience wind farm effects. The section concludes with some results of simulated drifter dispersion (Section 3.4).*"

This will provide a logical structure to the paper, and a setting that will allow testing working hypothesis. It might be that the data set is too small and the errors are too big to actually differentiate between conditional statistics, but at least this will be shown in a quantitative way. In the present version of the paper, the authors actually take a similar approach for the discussion of the spectra and structure functions, but the hypotheses are not presented in a clear fashion and are not consistent throughout the paper.

As the referee already states, strict hypothesis testing will not be possible given the small number of drifters and the variety of uncontrolled influencing factors. We cannot see how a statistical formalism could help overcome this very obvious fact. A major problem is also that distances between drifters and wind farms are ill-defined parameters. Given the size of the wind farm, it is not clear how such distances should be measured. Effective distances might also depend on wind direction relative to a drifter's location. It is possibly not very beneficial to apply formalized statistics to a small number of values that are just vaguely defined.

DETAILED COMMENTS

- Section 1
  - Lines 1-5. There are a number of recent papers that investigate "local" initial conditions (e.g. Ohlman et al, 2017; Berta et al., 2016; Poje et al., 2014)
    These three references were already referred to later in the paper. But we agree that they should be mentioned already here in the introduction. We changed the passage accordingly: "*The sub-mesoscale we focus on has*

*also been addressed by numerous other studies (e.g. Berta et al. 2016; Ohlmann et al. 2017; Poje et al. 2014). A key observation is that spreading rates may be much higher than those observed on the large-scale (Corrado et al. 2017).*"

- o Line 20 Please expand on the mechanisms through which OWF are expected to impact on surface dispersion
  The third paragraph of Section 1 (Introduction) has been revised, extending the already existing summary of relevant processes (wakes, vertical mixing, atmospheric or marine turbulence).

- Section 2
  - o Lines 10-20. Please discuss expected slippage errors of the MDO3 drifters. Have they been quantitatively tested? and compared with other types of drifters such as the classic CODE? Please provide references
  Slippage errors are now addressed in a new paragraph (second paragraph of Section 2.1). "*Although Albatros MD03 drifters have been widely used during the last years (e.g. Lana et al. (2016), Callies et al. (2017), Sentchev et al. (2017), Onken et al. (2018)), to our knowledge slippage of this drifter type has never been quantified. However, considering the drag ratio of 33.2, the parametrization exposed in Niiler et al. (1995) would predict a slippage of 1.1 to 1.6 cm/s, for 10 m/s wind speed and a velocity difference across the vertical extent of the drogue of roughly 0.1 cm/s. Quantification of a drifter's slip is not trivial due to an influence of sea-state. For another type of drifter, the {CODE} drifter, Poulain et al. (2009) estimated slippage to be 1 % of wind speed. By contrast, according to Poulain and Gerin (2019) slippage was estimated to be 0.1 % of wind speed. Fortunately, specification of slippage effects is of minor importance for the present study. First, it can be expected that slippage effects affecting two drifters of the same type will not dominate separation of these drifters. Second, when comparing observations with corresponding simulations, the additional wind drag tuned for successful simulations will cover also slippage effects. Therefore, for the present study slippage effects were neglected.*"
  Unfortunately drifter specific estimates are not available. However, it seems plausible that slippage effects will not dominate separation of identical drifters exposed to the same forcing. The wind drag assumed for numerical simulations will implicitly cover also slippage effects without, however, distinguishing them from effects of Stokes drift, for instance.

  - o Table 1. It should be improved or complemented by an other table. Initial distances between pairs and distances from OWFs should be included.
  We see the point that spatial scales of drifter separation should be indicated more clearly. To solve this problem, we added in each plot of drift trajectories (panels in Figs. 2, 4 and 6) an explicit length scale, which in particular emphasizes the small initial distances between drifters (< 100m).

  In Section 2.1 (after the description of the three drifter sets) we clearly state that initial drifter separations shown refer to the time at which the first signals were received from the positioning system. That means that initial separations are even smaller than shown, unfortunately the precise values cannot be specified.
  The referee would like to see information on initial distances from wind farms being included in a table. We thought about this idea but came to the conclusion that such information cannot be given in a meaningful way. Figures like Fig. 1a, for instance, show that the distance in question is

much smaller than the size of the wind farm. This means that it would rather arbitrary choice how to define the reference location of the wind farm. Should it be the location of the nearest engine or instead the centre of the wind farm? This choice would dominate the value one obtains. Therefore we came to the conclusion that a pure listing of such fuzzy numerical values would not be helpful for the reader, given the fact that the information the referee asks for is easily accessible from the trajectory plots in Figs. 2, 4 and 6.

- o Also in the text, in Section 2 and 3, please be more quantitative, avoid mentioning that pair are "close" or far, and refer to the i.c. in Table 1.
  We presume that this remark addresses mainly the discussion of Figs. 11-13 in Section 3.3 where we classified drifters with regard to their location relative to wind parks. As already mentioned, giving absolute distances is difficult as these are time dependent and wind farms cover large areas. However, the group of drifters being close to wind farms can also be described as those that even entered the wind farm area. Throughout the paper we now use this more precise wording.

- o Section 2.4. Please specify model initial distances between pairs and comment on the fact that given a model resolution of 900 m, local structures beyond 2-4 km are not correctly resolved.
  To simulate drifter dispersion, all particles are started at exactly the same location. This is said in the caption of Fig. 14 ("…*100 trajectories initialized at the same location…*") and also the first paragraph of Section 3.4 ("…*spreading from a common source point…*"). In the revised manuscript we now also included in Section 2.4 the following sentences, which explicitly address the problem of lacking grid resolution and stress the point that no initial particle separation is needed for simulating dispersion: "*Grid resolution limits the scale of flow features that can be resolved. Drifter separations of less than 1~km are clearly beyond the resolution of BSHcmod. The general approach to overcome such problem is to include sub-grid scale turbulent processes via a scale-dependent random diffusion term. With such approach being implemented, even particles released at the same initial location will start separating.*"

- o Fig. 1. It should be improved, showing the deployment design and the topography
  Thank you for giving this hint: Although in Fig. 1 the bathymetry was already shown, the numeric scale corresponding with the different colours was missing. In the revised manuscript, a corresponding legend has been added to the figure. We also found that in the horizontal length scale an error had slipped in. This has been corrected.

  Fig. 1 is meant to give an overview of the larger region where wind farms and corresponding drifter experiments are situated. At the spatial scale of Fig. 1 it is impossible to display the deployment design of the small scale drifter experiments. However, Fig. 1 clearly indicates the locations of the two wind farms within the German Bight region. Throughout the paper, each plot of drifter trajectories (such as Fig. 2a, for instance) shows these farms in much larger resolution. In our opinion each of these detailed plots, resolving even individual wind engines, displays very clearly how the respective drifters were deployed relative to the wind farm.

- Section 3
  - o Fig.3 5,7 and related text. The exponential fit seems very arbitrary to me. Were other fits tested as well? The initial distances from which the fit start

should be mentioned. Please discuss errors and confidence limits. In order to compare results, the initial distance should be comparable. See also the point on model pairs above. In general, please see General Comment above.

In an earlier version of the manuscript we also provided a fitted power law. However, these fits are very sensitive and obviously do not provide better results than the exponential fit. The following figure shows this for the example of Fig. 3 (see additional dotted lines). We therefore decided to not include this in the paper.

[Figure]

Panel (c) of the above figure also provides an example of how the data the exponential law is fitted to do not just correspond to a signal superimposed by some (e.g. Gaussian) noise. Roughly between 22 May 12:00 and 23 May 12:00 the fluctuations of squared drifter distance do not seem to be purely random. Therefore the exponential model is just a possibly weak indicator that underlying processes are not too far from theoretical expectations. Specification of uncertainties is not really meaningful or even possible in such context. We now comment on this problem at the end of Section 3.1.1: "*In sum, the exponential model should be seen as just an indicator of what could be expected theoretically. Specification of uncertainty bounds of the fitted model does not seem reasonable in this context.*"

o Section 3.2. The computed spectra are in time, while the general discussion in 2.2 is in terms of wavenumbers. Please discuss the hypotheses used to link the two types of spectra. The drifter spectra (except for one case) are obtained from time series of 1-3 days. Can they effectively resolve tidal frequency, even using MMT? Please discuss errors and confidence limits.

In Fig. 9 we considered energy as a function of frequency as this is the natural approach for the analysis of local time series. A transformation into the domain of wave numbers would have to be based on the assumption of some transport velocity. Panels in Fig. 10 are thought to be directly contrasted with Fig. 9 so that changing the independent variable would not make sense. An important aspect in the section is to identify the relevance of tidal motions. The most straightforward approach for doing that is an analysis in terms of frequencies, needing no further assumptions.

As suggested by the referee, we checked statistical significance of spectral peaks, a corresponding paragraph added at the end of Section 2.2 mentions the methods applied and gives all relevant references: "*Besides all mentioned advantages, a drawback of the MEM method is that the statistical significance of the spectral peaks is difficult to assess. Nevertheless, to estimate the statistical significance of spectral peaks the method applying a permutation test (Good, 2000) as proposed and exemplified by Pardo-Igúzquiza and Rodríguez-Tovar (2005, 2006) has been followed. Identified spectral peaks referred to in the discussion section show high statistical confidence levels with values between 95% and 99% based on the permutation test (10,000 spectra) using an underlying red noise spectrum.*" However, it is also to be noted that the tidal constituents indicated in Figs. 9 and 10 (magenta coloured lines) were not analysed from the data. They rather represent the values that are expected according to physics.

○ Section 3.3. What do the authors mean by "Eulerian and Lagrangian" separation?
The corresponding explanation has probably been a bit too short. We added (third paragraph of Section 3.3) the exact definition of the Lagrangian velocity increments: "*Increments $\delta v^{(L)}(t)$ were obtained as differences between velocities of the same drifter at times t and t+τ, where τ=20 min corresponds with the time resolution of drifter observations.*" Regarding Eulerian velocity increments we now explicitly refer to the definitions given in Eqs. (3) and (4).

○ Section 3.4. What are the initial distances of the model pairs? Given the model resolution, the dynamics is not expected to be local beyond 2-4 km, so that the exponential behavior is simply a consequence of the setting.
The referee is absolutely right, the exponential growth of distances is to be expected when this kind of parameterization is used in numeral modelling. Fig. 14 was included to demonstrate that. Initial distances between particles were assumed to be zero, stated in the figure caption: (" … *100 trajectories initialized at the same location …*")  and at the beginning of Section 3.4 *("… spreading from a common source point …").*

[revised manuscript text omitted]

---

## Author Response (AR2)

**Comments of the Referee:**

The revised version of the paper does not address any of my main criticisms and comments, as detailed in the following.

The authors choose not to perform any statistical test, "since it would generate a substantial formal overhead without promising a clear benefit". My point is that testing should be done, rather than avoided, and the importance of the parameters should be tested rather than speculated.

Our point is that any testing will necessarily be based on assumptions that obviously are not satisfied in the present case. Looking at Fig. 3, for instance, reveals the presence of tidal signals (mentioned in the paper) in observed dispersion, undermining the assumption of iid random fluctuations. Another problem is that uncertainties of GPS based localization can hardly be quantified. Finally, it must be kept in mind that graphs like those in Fig. 3 refer to just one drifter pair rather than averages over several pairs. For all these reasons we never claimed that we could really prove exponential growth.

Please see our more specific responses below.

They did not provide the requested table quantifying initial distances between pairs and from OWFS, arguing that they are not defined in a clear way. I am not sure what do they mean...As a simple example, the "initial distance" in most papers is simply defined as the distance corresponding to the first data points...

The referee addresses two different questions. The first is the specification of initial distances of drifter pairs. In the experiments, drifter pairs were released as closely as possible, but the only quantitative measure we can provide is the first GPS based estimate. This estimate may refer to a time approximately 20 minutes after drifter deployment, sometimes (see in particular drifters $C_9$ and $C_{10}$) technical problems produced an even much larger delay. In the revised manuscript, we now introduced a new table (Table 2), which provides these GPS based estimates of initial distances for all drifter pairs. The new table contributes to the discussion of observational uncertainty (Section 2.1).

The second point is the specification of initial distances between drifters and neighbouring wind farms. We obviously have not been clear in our previous response. We now introduced the table of relative distances the referee wants to see (Table 3) at the beginning of Section 3. However, our point is that values for the distances between drifters and wind farms depend very much on the choice made regarding what should represent the wind farm location. In Table 3 we provide two different measures, one referring to the wind farm centre and the other referring to the location of the nearest wind turbine. Differences between these two numbers may be large and a ranking regarding distance from the wind farm therefore somehow arbitrary. For instance: for drifter $A_1$ we get either 1.82 or 7.91 km, for drifter $B_1$ we get 4.10 or 7.28 km. From the nearest engine perspective, $A_1$ would definitely be closer, otherwise $B_1$ would be the closest to the wind farm. One option might be to choose always the nearest turbine option, assuming that drifters will always be within this turbine's turbulent wake. However, this situation is unlikely and the assumption cannot be substantiated based on observations.

They decided not to discuss and test the exponential fit. Their motivation is that there is no clear difference between results obtained using an exponential fit or other power law fits. But this is exactly my point! The exponential fit is simply not significant... And despite this, the authors mention in the abstract that "Drifter pairs can be classified in a remarkably clear way into those with spatial separation growing exponentially (and those growing) non-monotonically".

We agree that some of our statements benefited from a revision. A major point is that a strict distinction between exponential and power law growth in time is difficult. To better address this point, we decided to include a small new figure (Fig. 15) that re-displays squared pair distances from experiment B (for all of them fitted e-folding times are very similar) now using, however, a logarithmic time axis appropriate for fitting a power law. The figure nicely shows the similarity between exponential and power law growth for longer drift times. However, it also illustrates how fitting the power law would be influenced by the choice of the lower bound of the time interval the fit is based on. This is crucial as uncertainty of drifter location can just be estimated.

However, the discrimination between exponential and power law growth must be distinguished from the identification of drifter pairs for which separation grows non-monotonically. It is quite obvious that neither an exponential nor a power law would fit the behaviour shown in Fig. 7e, for instance. The key question is, whether it is just a coincidence that all the (three) cases with non-monotonic growth of drifter separation occur for exactly those drifters that were furthest from the wind farm (see the new Table 3). Unfortunately, the number of drifters is too small for a reliable answer to this question. The data provide, however, indications that motivate future more comprehensive studies.

We checked the manuscript carefully and revised all sentences that might be (mis)understood in the sense that we claim to really prove a hypothesis (including changes in the abstract and the conclusion). It is quite clear that 12 pairs of drifters released at different places under different atmospheric conditions (!) can not be the basis for a formalized hypothesis testing, developed for well defined experimental set-ups from which a number of independent realizations can be obtained. What is more, the expectation of exponential (or power law) growth of pair separation is based on theoretical models that try to predict dominant aspects of drifter behaviour in the presence of turbulence. The effects of simplifications in such theories can hardly be quantified. For our study this means that we are looking for some resemblance with predicted exponential separation rate, not being able to exactly quantify 'resemblance'.

For the reasons mentioned, qualitative assessment of particle pair separations is a common approach in the literature (Haza et al. 2008, Koszalka et al. 2009, LaCasce 2010, Berti et al., 2011). Hypothesis testing in this context is extremely difficult or even impossible and results on relative dispersion at submesoscale are still inconclusive.

Overall, all my reservations on this paper remain the same, and i cannot accept it for publication in the present form. As already stated, the paper is nicely written and the authors are knowledgeable, but i think the results are not clear nor robust enough, and they do not add significantly to our understanding of the problem.

In our paper we invoke a large number of other studies that report different experiments. Few of these experiments are conclusive when seen individually. Referring to large sets of data, Poje et al. (2014, 2017) or Corrado et al. (2017) try to summarize the state of knowledge. In a broader context the contribution from our study is that the e-folding times we fitted to the data agree with what was found also in other experiments (e.g. Koszalka, 2009; Schroeder, 2011; Corrado et al., 2017). Conversely, the results published in other studies to some degree support our approach.

We hope that Tables 2, 3 and Fig. 15 provide the information the referee was asking for. In particular, we hope that we were able to better discriminate between non-monotonic and non-exponential (i.e. power-law) growth of pair separation. This distinction is essential for our response to the criticism raised by the referee. The non-monotonic behaviour of three drifter pairs seems quite obvious, no meaningful theoretical model of turbulent dispersion could be fitted to these time series (Fig. 7). For all other drifters, the much more subtle statistical distinction between exponential and power law

behaviour (new Fig. 15) would need more comprehensive data. Our review of published experiments partially compensated for that limitation.

**Editorial comments:**

Abstract (page 1) line 3.  ". . non-monotonically. There . ."
Has been changed.

Section 2.1 page 4
Line 17.  "exposed" should be "expounded" or "given" or perhaps simply omitted?
Has been replaced by 'given'.

Line 31.  ". . study on the extent to which position errors . ."
Has been corrected.

Section 2.2 page 6 line 13.  Simpler ". . significance of spectral peaks, a permutation"
Has been changed accordingly.

Section 2.4 page 9 line 6.  Better ". . This value, estimated by Callies et al. (2017) for the same drifter types, seems largely consistent . ."
Has been changed.

Section 3.1.3 page 18 lines 3-6 and figure 8.  You do not show the squared distances for all possible drifter combinations as you do not involve C2, C6, C8, C10 (nor C3, presumably because too short).  I guess that this is because the separation between C1 and C2 (for example) would make little difference to the squared distance plot with C4, C5, C7, C9.  However, in view of the referee's concern about the small numbers of drifters, I think you should at least comment on this.
We agree and added the following explanation: *"Due to the fact that initial distances between drifters released together are negligible relative to distances between different drifter pairs, we involved just one drifter from each pair (we chose C4 over C3 as the latter drifter's trajectory ended early and had little temporal overlap with other drifters, see Fig. 6c)"*

Section 3.3 page 21.
Line 6.  ". . locations (as occurs . ."
Has been corrected

Line 9.  "direct contact with OWFs".  Please state a definition or criterion for this, as used to distinguish the plots in figures 11, 12.  (Also a referee concern).

[revised manuscript text omitted]

---

## Author Response (AR3)

**Topic Editor Decision: Publish subject to technical corrections** (07 Jun 2019) by John M. Huthnance
Comments to the Author:
Dear Authors
Thank-you for your re-revised manuscript. I have now discovered that my interspersed comments on the referee comments were unfortunately somehow omitted from my message to you via the Editorial system. I am including them below with different annotation to distinguish what the referee commented (you saw that) from my comments thereon.
Anyway, I think that you have mostly responded in accord with what I suggested. But you should note my comments marked **. In respect of the second and third of these, I suggest that you might say a bit more at the end about how to design a follow-up experiment that could give results with a desired confidence level. Your conclusion is at present a bit vague about this.
Please also note "Detailed comments" at the end of what follows.
I am regarding these as "Technical Corrections" meaning that you should then upload your manuscript to the Copernicus / Ocean Science editorial system directly (no more intervention by myself). There will be copy editing and you should check that your intended meaning is retained.
Thank-you for publishing in Ocean Science.
Yours sincerely
John Huthnance

**Answer:** We followed your advice and provided more specific ideas for future experiments in the last paragraph of the conclusions section.

Referee
"The revised version of the paper does not address any of my main criticisms and comments, as detailed in the following."
Editor.
I think that most of these criticisms and comments are reasonable and should be addressed. When/if the manuscript is published, the referee comments will be available and readers will be able to see if they have been addressed reasonably.**

**Answer:** In the second iteration of the revision of the manuscript we added the tables the referee wanted to see. We also introduced another figure (Fig. 15) and revised the text in several places.

Referee
"The authors choose not to perform any statistical test, "since it would generate a substantial formal overhead without promising a clear benefit". My point is that testing should be done, rather than avoided, and the importance of the parameters should be tested rather than speculated."
Editor.
The benefit is better evidence about the "robustness" of the results, and possibly how many drifters in a follow-up experiment might be enough to give results with a desired confidence level.**

**Answer:** In our previous response, we addressed the problem that also the theoretical considerations behind the expectation of exponential growth, for instance, include simplifying assumptions. Uncertainties arising from these assumptions can hardly be quantified, which makes strict testing a questionable endeavour. The other, even more severe, problem is that the collection of data we

analysed did not result from different realizations within a well-defined experimental setup. During the three experiments we combined, weather conditions were quite different.

Referee.
"They did not provide the requested table quantifying initial distances between pairs and from OWFS, arguing that they are not defined in a clear way. I am not sure what do they mean…As a simple example, the "initial distance" in most papers is simply defined as the distance corresponding to the first data points…"
Editor.
I am sure you can choose your own reasonable definition for initial distances between pairs, e.g. as suggested. Obviously there is no one point for OWF location; however, shortest distance to a turbine pylon might be a meaningful measure for distance to an OWF – the choice can be yours as long as it is defined.

**Answer:** In the new Table 3, the nearest engine version of distance has been contrasted with an estimate based on the distance from the wind farm centre. Differences between the two approaches are substantial and ranking drifter pairs with regard to distance from the wind farm is not conclusive.

Referee
"They decided not to discuss and test the exponential fit. Their motivation is that there is no clear difference between results obtained using an exponential fit or other power law fits. But this is exactly my point!"
Editor.
I have to say that your text gives this comment much validity. The introduction, page 2 lines 31-33 says "Indicative of a non-local regime driven by flow features larger than drifter separation is exponential growth of relative drifter dispersion (LaCasce, 2008). By contrast, local dispersion with power law dependence on time should coincide with a shallower slope of the energy spectrum, indicating the presence of energetic small scale eddies." This implies that you want to distinguish between exponential and power-law time-dependence in order to distinguish between large flow features (or Lagrangian chaos, page 3 line 3) and small-scale eddies in causing dispersion.

**Answer:** The new Fig. 15 is now dedicated to the problem of distinguishing between exponential and power law fits. See our previous responses to the referee's comments.

Referee
"The exponential fit is simply not significant…"
Editor.
I think this statement needs qualifying. I guess that, assuming an exponential form, the exponent is very significantly positive. However, it might well be that a quadratic form could fit equally well and in view of the above you should try this.

**Answer:** We found estimates of power law exponents being more unstable than those of e-folding times in the exponential fit (we now mention that at the end of the third paragraph of Section 4.1).

See, however, Fig. 15 for a comparison of the exponential fit (non-local dispersion) with an increase as t^3, as it would be expected according to Richardson's law for local dispersion.

Referee
"And despite this, the authors mention in the abstract that "Drifter pairs can be classified in a remarkably clear way into those with spatial separation growing exponentially (and those growing) non-monotonically"."
Editor.
This is another matter! It seems to me that it is much easier to distinguish between exponential and non-monotonic behaviour than between exponential and quadratic. I think your non-monotonic plots 7a,b,d,e are indeed clearly different from those where you fit an exponential. An exponential cannot fit the overall convex shape. (A quadratic could but of course with the opposite (wrong?) sign of the t^2 term). So you could refute this referee comment.

**Answer:** We think that we now discussed the distinction between non-exponential (i.e. power law) and non-monotonic growth behaviour in a proper way.

Referee
"Overall, all my reservations on this paper remain the same, and i cannot accept it for publication in the present form. As already stated, the paper is nicely written and the authors are knowledgeable, but i think the results are not clear nor robust enough, and they do not add significantly to our understanding of the problem"
Editor.
I am not rejecting it at this stage, but you should please address the comments in the light of what I say above. This assessment of results seems fair, apart from your clear distinction between exponential and non-monotonic separation behaviour. Perhaps you might be a bit more specific about how future work might enable attribution of the different behaviours to context.**

**Answer:** In addition to our previous responses, we now outlined a more specific concept for future studies at the end of the conclusions section.

Editorial comments . . . . . . [You saw all of this and responded].

Detailed comments.
Page 1 line 16 and page 3 line 7. Better "early-phase"? (with hyphen to avoid suggesting phase separation)
Page 13 line 8. Omit "in the"
Page 23 line 5. Better ". . increase approximately as r2/3, as expected . ."
Page 28 lines 3-4. Better ". . (GDP), Corrado . ."
Page 32 line 16. "are also" -> "also include"?

**Answer:** All the above changes have been made.